# Graph drawing using Jaya

**Fadi K. Dib** [1]*, **Peter Rodgers** [2]

**1** Computer Science Department, Center for Applied Mathematics and Bioinformatics (CAMB), Gulf University for Science and Technology (GUST), Hawally, Kuwait, **2** School of Computing, University of Kent, Canterbury, Kent, United Kingdom

* deeb.f@gust.edu.kw

## Abstract

Graph drawing, involving the automatic layout of graphs, is vital for clear data visualization and interpretation but poses challenges due to the optimization of a multi-metric objective function, an area where current search-based methods seek improvement. In this paper, we investigate the performance of Jaya algorithm for automatic graph layout with straight lines. Jaya algorithm has not been previously used in the field of graph drawing. Unlike most population-based methods, Jaya algorithm is a parameter-less algorithm in that it requires no algorithm-specific control parameters and only population size and number of iterations need to be specified, which makes it easy for researchers to apply in the field. To improve Jaya algorithm's performance, we applied Latin Hypercube Sampling to initialize the population of individuals so that they widely cover the search space. We developed a visualization tool that simplifies the integration of search methods, allowing for easy performance testing of algorithms on graphs with weighted aesthetic metrics. We benchmarked the Jaya algorithm and its enhanced version against Hill Climbing and Simulated Annealing, commonly used graph-drawing search algorithms which have a limited number of parameters, to demonstrate Jaya algorithm's effectiveness in the field. We conducted experiments on synthetic datasets with varying numbers of nodes and edges using the Erdős–Rényi model and real-world graph datasets and evaluated the quality of the generated layouts, and the performance of the methods based on number of function evaluations. We also conducted a scalability experiment on Jaya algorithm to evaluate its ability to handle large-scale graphs. Our results showed that Jaya algorithm significantly outperforms Hill Climbing and Simulated Annealing in terms of the quality of the generated graph layouts and the speed at which the layouts were produced. Using improved population sampling generated better layouts compared to the original Jaya algorithm using the same number of function evaluations. Moreover, Jaya algorithm was able to draw layouts for graphs with 500 nodes in a reasonable time.

**Data Availability Statement:** All relevant data for this study is publicly available in the OSF repository (https://doi.org/10.17605/OSF.IO/6AFVR).

**Funding:** The author(s) received no specific funding for this work.

## 1. Introduction

Graph drawing deals with the visualization of complex networks, such as social networks, biological pathways, and communication networks. Automatic graph layout aims to generate an aesthetically pleasing and easy-to-understand layout for graphs by optimizing specific quality

**Competing interests:** The authors have declared that no competing interests exist.

metrics [1]. It involves arranging nodes and edges of a graph in a meaningful way. A well-drawn graph should have a clear layout that facilitates interpretation and analysis. Graph layout is essential in various fields, including computer science, engineering, biology, and social sciences, where effective visualization of network data is vital.

Graph layout quality is typically evaluated using several metrics, such as edge crossings, edge length, node overlap, angle resolution, and aesthetic criteria such as symmetry [2, 3]. These metrics can be combined into a multi-objective function and optimized using various methods.

While other methods, such as dimension reduction and multi-level techniques, have been used for graph drawing, the majority of methods can be broadly categorized into force-directed methods and search-based methods for drawing graphs with multiple metrics [4]. Force-directed methods mimic physical systems, where the edges of the graph represent springs, and nodes are electrically charged particles. On the other hand, search-based methods search the solution space by generating a sequence of candidate solutions and iteratively improving them based on a predefined objective function. Search-based methods have gained popularity in graph drawing due to their ability to handle multi-objective functions. These methods can combine multiple quality metrics into a single function to be optimized.

Researchers in the field of multi-criteria graph drawing benefit from methods that are easy to implement and which require few parameters to tune. This allows researchers and practitioners to focus on solving the graph layout problem rather than spending significant time and effort fine-tuning algorithm parameters [5]. Hill Climbing and Simulated Annealing are popular optimization techniques in the field of graph drawing that require fewer parameters to tune compared to other optimization techniques. Hill Climbing is a simple and effective optimization algorithm that examines neighboring solutions and moves to the best one until a local optimum is reached [6], while Simulated Annealing can escape local optima by allowing for worse solutions to be accepted with a decreasing probability as the algorithm progresses [2]. While these methods have been proven effective for graph layout optimization and their simplicity make them accessible and user-friendly, they have great potential for improvement. For example, Hill Climbing is generally faster in reaching a final layout, but the final result is not always the best as it is more likely to get trapped in a local optima, while Simulated Annealing adds an element of non-determinism to escape from local optima, which slows down the performance of the algorithm [7]. Both are neighborhood search methods which explore only from a single current solution, so restricting the amount of search space they consider.

In this research, we propose the Jaya algorithm [8] for automatic graph layout to improve the efficiency and effectiveness of drawing general graph layouts with straight lines based on a weighted sum multi-criteria optimization. To our knowledge, Jaya algorithm has never been used in the field of graph drawing. Jaya algorithm is a population-based search method that maintains a population of candidate solutions, where solutions are updated based on the best solutions found in the population. We introduce Jaya algorithm as it has no algorithm-specific control parameters other than population size and number of iterations, which makes it easy for researchers to apply in the field, and it has been proven effective in many applications [9, 10].

Our study aimed to answer two main research questions. Firstly, does Jaya algorithm perform better than Hill Climbing and Simulated Annealing approaches in the field of graph layout? Secondly, does applying Latin Hypercube Sampling (LHS) for initializing the population for Jaya algorithm improve the performance of the original Jaya graph drawing algorithm? To answer these questions, we implemented and evaluated these methods alongside Hill Climbing and Simulated Annealing. We conducted three types of evaluations: finding

the best layout achievable by minimizing the value of the objective function, measuring the quality of the graph layout after a fixed optimization time (number of function evaluations), and determining the speed to draw an acceptable layout. We compared Jaya algorithm with Hill Climbing and Simulated Annealing, as they have fewer parameters to tune than other search methods.

The research in this paper contributes to the field of graph layout optimization by proposing a new method, Jaya, and evaluating its performance against well-established optimization methods. The results demonstrate that Jaya algorithm can produce better results than Hill Climbing and Simulated Annealing, highlighting its potential in solving complex graph layout problems. The addition of LHS for population initialization within the proposed Jaya procedure speeds up the identification of good solutions and outperforms the original Jaya algorithm by producing graph layouts with better values of objective function. The code and data related to this research can be accessed from Open Science Framework at https://doi.org/10.17605/OSF.IO/6AFVR.

The main contributions of this research can be summarized as follows:

- Our study pioneers the use of the Jaya algorithm in graph drawing, outperforming conventional search optimization methods significantly. It is a parameter-less algorithm, requiring no algorithm-specific control parameters, making it easy for researchers to apply in the field.

- We optimized the Jaya algorithm by integrating the Latin Hypercube Sampling (LHS) method for population initialization, boosting its overall efficiency.

- We developed an intuitive visualization tool that facilitates the evaluation and comparison of different optimization techniques for graph layout by researchers and practitioners.

The remainder of this paper is structured as follows: In Section 2, the background of graph drawing, search-based techniques, and Jaya algorithm are explained. Section 3 introduces our proposed Jaya algorithm and details the addition of Latin Hypercube Sampling (LHS) during the initialization of population. Section 4 covers the experimental results, including the visualization tool description, the parameter tuning process, the experiment outcomes of applying Hill Climbing, Simulated Annealing, Jaya algorithm, and Jaya algorithm with LHS on both random and real-world datasets, and the scalability testing of our approach. In Section 5, the results are discussed and analyzed. Finally, Section 6 presents the conclusions and suggests potential areas of future work.

## 2. Background

Graph drawing is the process of visually representing graphs, which consist of nodes (vertices) and edges (links) that connect them. The goal of graph drawing is to create a layout that effectively communicates the structure of the graph to the viewer. However, designing a good layout is challenging as it involves balancing graph aesthetics with efficiency of the automated implementation [11]. Graph aesthetics are quality measures that determine the readability and usability of graphs. A good layout should provide clear and meaningful information, whereas a poor layout can be confusing and misleading [3]. The metrics used in graph drawing aim to create aesthetically pleasing layouts that improve the readability of the graph and enable the viewer to easily perceive the topological structure of the graph [12]. To evaluate the quality of a graph layout, an objective function comprising metrics in a weighted sum can be used. The objective function might include measures such as minimizing edge crossings, minimizing edge bends, uniform edge length, maximizing graphs symmetry, maximizing node-to-node and node-to-edge occlusions, and maximizing angular resolution of incident edges. However,

computing the objective function for every layout examined can be computationally expensive, as some metrics can be time-consuming to calculate, and the overall objective function may include both continuous and discrete functions [13]. To address this challenge, search-based optimization methods such as Simulated Annealing, Hill Climbing, and Genetic Algorithms have been used to find a good value of the objective function.

## 2.1. Search-based optimization methods for graph drawing

Simulated Annealing emerged as one of the initial search-based techniques applied to graph layout problems [2], facilitating the visualization of graphs with straight edges. Although the original algorithm yields satisfactory results for smaller graphs, its performance declines with larger graphs. This method emulates the process of heating a material and gradually reducing the temperature to minimize imperfections, consequently lowering the system's energy. By incorporating uphill moves with diminishing probability throughout the search, the approach aims to transition from local minima to global minima.

In [14] Simulated Annealing was used to solve the problem of bipartite graph crossing minimization aiming to minimize edge crossings in a bipartite graph with parts embedded along two parallel lines. The proposed approach combines a Simulated Annealing method with a Variable Neighborhood Search scheme, executed iteratively.

Additionally, Simulated Annealing was employed to tackle the challenge of creating map-like visualizations, intending to foster cognitive mapping between the map vehicle and the target ontology through psychological association [15]. The authors suggest using a Simulated Annealing algorithm to guide the decision-making process for shape determination, thereby enhancing mapping efficiency. Experiments conducted on real data reveal that this method not only generates a comprehensive outline closely aligned with the predefined shape but also achieves internal regions characterized by simplistic shapes.

Hill Climbing represents an alternative search-based method employed within the domain of graph visualization, aiming to reduce edge intersections [6]. This technique offers a more straightforward and accelerated search process compared to Simulated Annealing, as it avoids making uphill moves. However, this characteristic often leads to suboptimal local solutions due to entrapment in local optima.

In [16] Hill Climbing was employed to draw area-proportional Euler diagrams with ellipses. It contributes to this work by providing a fast and effective method for solving the problem, albeit with the limitation of becoming stuck in local optima.

Genetic Algorithm, a computational approach inspired by the process of natural selection, was proposed in [17] for generating graph representations while considering numerous visual constraints. This method can produce good graph layouts. Despite its effectiveness, the primary drawback of this approach is its slow rate of convergence. Genetic algorithms also require significant parameter tuning, including crossover and mutation probability.

Also, the work in [18] proposes GenMap, an application mapping framework using multi-objective optimization based on a genetic algorithm. It provides aggressive power optimization using a dynamic power model and leakage minimization technique to optimize spatial mapping of coarse-grained reconfigurable architectures.

Previous research has explored other search-based techniques, including our own work on Tabu Search and Path Relinking for graph drawing optimization [19]. However, the parameters tuning process can be challenging for such methods, as the proposed algorithm required 11 parameters, 5 for Tabu Search and 6 for Path Relinking, which made it less suitable for our current study.

## 2.2. The Jaya algorithm and the importance of parameter tuning

The work in [20] addressed the challenge of parameter tuning as a key issue in optimizing algorithms. It is important to find the best parameter settings for a given algorithm to achieve its best performance for a given set of problems. However, this can be a time-consuming process, and there is no guarantee that a well-tuned algorithm will work well for all types of problems. Additionally, even if an algorithm is tuned, its parameters become fixed after tuning. One way to address this challenge is to consider parameter tuning as a bi-objective process to form a self-tuning framework, where the algorithm to be tuned can be used to tune itself.

Researchers often gravitate towards optimization techniques that possess a minimal number of parameters, as this allows them to concentrate on addressing the problem at hand rather than devoting considerable time and effort to fine-tuning algorithm parameters. Employing optimization methods with fewer parameters simplifies the process, reduces the risk of overfitting, and generally leads to more efficient and reliable results. By minimizing the complexity associated with parameter tuning, researchers can dedicate their resources to solving the core problem and refining the solution [5].

The Jaya algorithm, proposed in [8], is presented as an optimization method with no algorithm-specific control parameters, making it simpler and more effective for solving optimization problems. It has demonstrated efficacy in various applications such as feature selection [21], image processing [22], planning and scheduling [23], as well as engineering contexts [24].

In [10], the pros and cons of the Jaya algorithm were addressed. Its main advantages include simplicity, user-friendliness, parameter-free design, derivative-free nature, and sound-and-complete attributes, which contribute to its success and widespread adoption. However, the Jaya algorithm faces several limitations, including its association with the No Free Lunch (NFL) Theorem of optimization, which means its convergence behavior depends on the nature of the problem's search space. Additionally, the algorithm's performance is influenced by the problem domain being addressed and its population-based behavior. While the algorithm can explore a vast range of search space regions through its unique operator, which evolves the search based on values attracted to the global best solution and moves closer to better areas in the search space by increasing the distance from worse solutions, it does not concentrate on each search space region to which it converges. This deficiency hampers the exploitation process, especially for problems with multi-modal landscapes.

## 2.3. Latin hypercube sampling in optimization algorithms

The initial population in optimization algorithms plays a crucial role in determining the performance and convergence of the optimization process. A diverse and well-distributed initial population can increase the exploration capability of the algorithm, facilitating a more effective search of the solution space, and potentially leading to higher quality solutions. On the other hand, a poorly chosen initial population may result in premature convergence or stagnation, causing the algorithm to become trapped in local optima [25].

Latin Hypercube Sampling (LHS), proposed in [26] is a statistical sampling method used in computer simulations and mathematical modeling to efficiently generate a representative sample of input parameter values from a multi-dimensional distribution. LHS stratifies each input variable independently into equal probability intervals, ensuring that each interval is sampled exactly once. This technique provides a more uniform coverage of the parameter space compared to random sampling, reducing the number of samples required for accurate estimates.

Employing Latin hypercube sampling in population-based multi-objective optimization has proven effective, as it guarantees thorough sampling of the design space and maintains

independence among design variables. This approach facilitates the generation of a diverse solution set and lowers the computational expense of the optimization process [27].

The current state of the art in search-based graph drawing methods have a number of drawbacks leading to a research gap. Techniques such as Simulated Annealing, Hill Climbing, and Genetic Algorithms often struggle with suboptimal solutions, slow rates of convergence, and the complexity of parameter tuning. Additionally, the initial population choice in can significantly impact the performance and convergence of the optimization process, potentially leading to premature convergence or stagnation. Although some research has addressed parameter tuning as a bi-objective process, and various methods have been proposed for efficient population initialization, such as Latin Hypercube Sampling (LHS), these techniques have not been combined in a comprehensive approach to graph drawing. Moreover, the potential of the Jaya algorithm remains unexplored in the field of graph drawing. This research aims to address these gaps by pioneering the use of the Jaya algorithm in graph drawing, integrating LHS for efficient population initialization, and facilitating the evaluation and comparison of different optimization techniques for graph layout through a user-friendly visualization tool.

## 3. Proposed method

This section describes our graph drawing method based on Jaya algorithm and how we improve it with Latin Hypercube Sampling (LHS) for initializing the population.

The proposed Jaya algorithm starts with generating a population of random candidate solutions (graph layouts), evaluates their quality based on an objective function, and iteratively improves the solutions by updating the coordinates of nodes.

The algorithm first generates an initial population of randomly generated candidate graph layouts. This method generates a population of *populationSize* individuals, each individual is a layout for the given graph. For each individual, random positions are generated for the nodes in the graph by using the screen bounds to ensure that the positions are within the visible area. The value of the objective function for each individual is calculated. Then, the best and worst layouts in the initial population are identified. The algorithm is demonstrated in Algorithm 1.

```
Algorithm 1. The Jaya algorithm for graph layout.
Algorithm Jaya (G, populationSize, iterations)
  Input: Graph G, populationSize, iterations
  Output: Best candidate
1: Initialize population of candidate graph layouts with random
positions
2: Evaluate cost of each candidate
3: Find best and worst candidates
4: For I from 1 to iterations do
5:    generateNewPopulation(population[], best, worst)
6:    Update best and worst candidates in population
7: End For
8: Return best candidate
```

The algorithm then enters a loop that runs for a predetermined number of *iterations*. In each iteration, the algorithm generates new candidate solutions using the `generateNewPopulation` method as shown in Algorithm 2. This method generates a new candidate for each population member. It updates its position by applying the Jaya algorithm update rules [8]. The updated candidate is then evaluated using the objective function, and the new candidate replaces the current candidate if it has a better cost.

```
Algorithm 2. Generate new population algorithm.
Algorithm generateNewPopulation(population[],bestCandidate,
worstCandidate)
  Input: population[], bestCandidate, worstCandidate
```

```
 Output: population[]
1: For i from 1 to populationSize do
2:    updateCandidateJaya(i, bestCandidate, worstCandidate)
3:    Evaluate cost of candidate
4:    If candidate.cost < population[i].cost then
5:      population[i] = candidate
6:    End If
7: End For
8: Return population[]
```

The `updateCandidateJaya` method, illustrated in Algorithm 3, applies the Jaya algorithm update rules to a given candidate. For each node, the method calculates a new position using the best and worst candidates from the current population. A new candidate graph is generated by calculating the position of each node in the graph. The position is calculated based on the current position of the node and the positions of the same node in the best and worst candidate graphs. The calculation involves adding a random value to the current position of the node. Additionally, two other random values are subtracted from the result. The first subtracted value is determined by the difference between the best candidate position and the current position of the node. This helps to move the node towards a better position in the graph. The second subtracted value is derived from the difference between the current position and the worst candidate position. This helps to move the node away from potentially unfavorable positions in the graph. By combining these three values, Jaya algorithm generates a new candidate position for the node. The algorithm keeps track of the best and worst candidates in the population. At the end of the algorithm, the best candidate is selected as the final solution.

**Algorithm 3. Update candidate algorithm.**

```
Algorithm updateCandidateJaya(candidate, bestCandidate,
worstCandidate)
  Input: candidate, bestCandidate, worstCandidate
  Output: candidate
1: For each node in candidate do
2:    xNew = candidate[node].x + rand() * (bestCandidate[node].x – abs
(candidate[node].x)) – rand() * (worstCandidate[node].x – abs(candi-
date[node].x))
3:    yNew = candidate[node].y + rand() * (bestCandidate[node].y – abs
(candidate[node].y)) – rand() * (worstCandidate[node].y – abs(candi-
date[node].y))
4:    candidate[node].x = xNew
5:    candidate[node].y = yNew
6: End For
7: Return candidate
```

Our proposed Jaya algorithm uses a conventional strategy for search-based graph drawing techniques that aim to optimize graph aesthetics. We evaluated the quality of the graph by employing four widely-used metrics in a weighted objective function, which have also been utilized in previous studies [2, 19, 28, 29]. These metrics measure the distribution of nodes, how similar the edge lengths are, edge crossings, and angular resolution.

We measure the distribution of nodes across the drawing space by maximizing the distances between nodes. Similar edge lengths are measured by specifying a target length and finding a sum of how far each edge is from this length. We count edge crossings, aiming to decrease that number in each optimization iteration. Finally, we improve angular resolution by measuring the distance between adjacent edges connected to a node and giving high values to small angles. The formulas utilized in our study are identical to those presented in [2, 29].

The objective of uniformly distributing the nodes across the drawing space necessitates maximizing the distances between nodes. This criterion was quantified using the subsequent

formula:

$$\sum_{i \in V} \sum_{j \in V} \frac{1}{d_{ij}^2} \tag{1}$$

where $d_{ij}$ represents the Euclidean distance between two nodes i and j, and $i \neq j$.

Equalizing the edge lengths involves designating a specific length (len), followed by adjusting all edges to conform to this predetermined length. This was accomplished using the formula as follows:

$$\sum_{e \in E} (e - len)^2 \tag{2}$$

where E is the set of edges.

For the minimization of crossing edges, our approach primarily involved identifying the number of intersecting edges, and endeavoring to reduce this count throughout each iteration of the optimization process.

Concerning the final metric, the angular resolution, we determined this by amplifying the separation between incident edges, accomplished through the application of the subsequent formula:

$$\sum_{v \in V} \sum_{\{e_1, e_2\} \in E} \left| \frac{2\pi}{\deg(v)} - \theta(e1, e2) \right| \tag{3}$$

where deg(v) denotes the degree of a node v, and

$\theta(e1, e2)$ is the angle in radians between two adjacent edges e1 and e2 incident to node v.

All these metrics contribute to the graph quality objective function, which is a weighted sum multi-criteria function computed as follows:

$$objectiveFunction = w_1 * m_1 + w_2 * m_2 + w_3 * m_3 + w_4 * m_4 \tag{4}$$

where $w_i$ and $m_i$ are the weight and the measure for the criterion i respectively.

We introduced a normalization process to ensure each measure's value ranges between 0 and 1 to avoid one measure overpowering the others. We assign equal weights to all criteria since it was challenging to establish universal weights that perform well for all graph sizes. We do not concentrate on generating the best set of weights for the criteria as our goal in this paper is to compare multi-objective methods for graph drawing, rather than produce the best possible layout. In any case the values for weights will depend on application area and user preference.

As with other population-based methods, the quality of the initial population significantly influences the Jaya algorithm's convergence rate and the quality of the obtained solutions [30]. In our improved Jaya implementation, we employ Latin Hypercube Sampling (LHS) for population initialization.

LHS [26] is a sampling technique utilized in diverse optimization problems. It functions by segmenting the search space into a grid of equal-sized subspaces and selecting a single point from each subspace. This method ensures that the sampled points are uniformly distributed throughout the search space, encompassing a broad range of potential solutions.

To apply LHS in the context of automatic graph drawing, we partition the plane, representing the search space, into four sections: upper left, upper right, lower left, and lower right. We then create five sets of individuals, with each set consisting of individuals. Each set is generated

by constraining the initial layout nodes to a specific section, such as upper left, upper right, lower left, lower right, or randomly dispersed throughout the search space.

This means that the initial population covers the search space comprehensively. By generating individuals in distinct areas, we minimize the probability of encountering an empty region in the resulting layout. Furthermore, by incorporating a set of randomly distributed individuals, we ensure thorough exploration of the search space, potentially leading to the discovery of superior solutions.

We chose to compare Jaya, a population-based graph layout method, with Hill Climbing and Simulated Annealing, as they are both widely-used and well-established neighborhood-based search techniques for graph layout optimization. Hill Climbing is a simple and effective optimization algorithm that examines neighboring solutions and moves to the best one until a local optimum is reached. In contrast, Simulated Annealing is another neighborhood-based method that can escape local optima by allowing for worse solutions to be accepted with a decreasing probability as the algorithm progresses.

The rationale behind this decision is to evaluate the performance of Jaya algorithm against search algorithms for graph drawing that also have few parameters to tune. We also aim to assess the ability of Jaya algorithm to adapt to the complexities of the graph layout problem, as well as its performance in terms of convergence rate, solution quality, and computational efficiency. By comparing these algorithms, we can demonstrate the effectiveness and efficiency of methods with low numbers of parameters, encouraging other researchers in the field to consider using such approaches. This comparison not only highlights their relative performance but also emphasizes the benefits of using algorithms that require less tuning, allowing practitioners to focus more on solving the graph layout problem instead of spending significant time and resources fine-tuning algorithm parameters.

Additionally, the reduced need for parameter tuning makes these methods more robust and generalizable across various graph layout problems. This can lead to more consistent performance and better reproducibility in research.

## 4. Experimental results

Our study aimed to answer two main research questions. Firstly, does Jaya algorithm perform better than Hill Climbing and Simulated Annealing approaches in the field of graph layout? Secondly, does applying Latin Hypercube Sampling (LHS) for initializing the population for Jaya algorithm improve the performance of the original Jaya graph drawing algorithm? To answer these questions, we implemented and evaluated these methods alongside Hill Climbing and Simulated Annealing, two well-known search algorithms frequently used in the field of graph drawing. Comparing Jaya algorithm with these methods provides a useful benchmark for evaluating its performance and potential advantages. As Jaya algorithm, Hill Climbing and Simulated Annealing have fewer parameters to tune than other search methods, we are comparing a group of low parameter search methods. We conducted three types of evaluations: finding the best layout achievable by minimizing the value of the objective function, measuring the quality of the graph layout after a fixed optimization time (number of function evaluations), and determining the speed to draw an acceptable layout. These evaluations allowed us to examine different possible use cases for graph layout. We compared the performance of Jaya algorithm and its improved version to Hill Climbing and Simulated Annealing after an initial investigation to find acceptable values for parameters.

In our implementation for Hill Climbing and Simulated Annealing, we employ a methodical approach to search for potential solutions by focusing on a square surrounding each node at a specific distance. This square consists of eight points, including the top, bottom, left, right,

and four corners. For each potential solution, we calculate the value of the objective function and choose the solution with the lowest value. If multiple solutions have the same value, we opt for the first one we encounter, starting from the rightmost solution and moving clockwise [19].

This technique is utilized in these two methods because they are based on the concept of neighborhood search, and it is necessary to have a clear definition of neighboring candidate solutions. As a result, we use a geometric shape to define the search space, which is in line with previous studies in graph drawing that employed a circle or rectangle [2, 29]. However, evaluating the multi-criteria objective function is time-intensive, so we restrict the search movements to only eight points to decrease execution time. Note that the square's size is a tunable parameter for each method. Although our implementation permits an increase in repetitions beyond eight points, this would substantially extend the execution times. In contrast to neighborhood search, Jaya is a population-based method, working with a group of candidate solutions simultaneously, rather than focusing on improving a single solution. Population based methods rely on the interactions among the individuals in the population, exploiting the diversity and the collective knowledge of the group to explore and exploit the search space more efficiently.

## 4.1. Visualization tool

MGDrawVis (Metaheuristics Graph Drawing Visualizer) is a Java-based visualization tool that facilitates the comparison of different search algorithms for automatic graph layout. The tool was developed using JavaFX for graphical libraries and Apache Netbeans 15 as an editor for Java. The code includes classes for each algorithm, with additional metaheuristics requiring implementation as a Java class. The code can be accessed from Open Science Framework at https://doi.org/10.17605/OSF.IO/6AFVR.

MGDrawVis offers a wide range of functionalities to users, including the ability to draw a graph by adding nodes and edges using mouse clicks or by uploading a file in Microsoft Excel Worksheet format. Users can also manually change the layout of the graph by dragging the nodes to the required positions, thus providing greater flexibility in choosing the initial layout. The tool can generate single or multiple randomly connected graphs using the Erdős–Rényi model, and it allows the user to save the layout for future use. Fig 1 shows a screenshot of MGDrawVis's interface.

One of the key features of MGDrawVis is its ability to work with a weighted objective function, which allows the user to change the weights of each metric, including desired distances such as node-node occlusion. The tool provides a button for each algorithm, and when a method is selected, the user can adjust the values of the algorithm's parameters. Once the layout optimization is complete, MGDrawVis displays the values of each metric along with the total value of the objective function, number of function evaluations, and execution time in seconds.

In addition, MGDrawVis supports batch file testing, allowing users to upload multiple files and run a chosen algorithm for three different scenarios: finding the best layout that can be generated by the method, finding the best layout that can be generated within a fixed number of function evaluations, or optimizing for a desired objective function value. The tool generates a Microsoft Excel Worksheet for each experiment, including the maximum, minimum, average, and median objective function values, number of function evaluations, and execution time in seconds for all runs. It also generates a PNG file for the output layout of each test case.

Overall, MGDrawVis provides a powerful and flexible tool for evaluating and comparing different algorithms for automatic graph layout. Its intuitive user interface and support for a

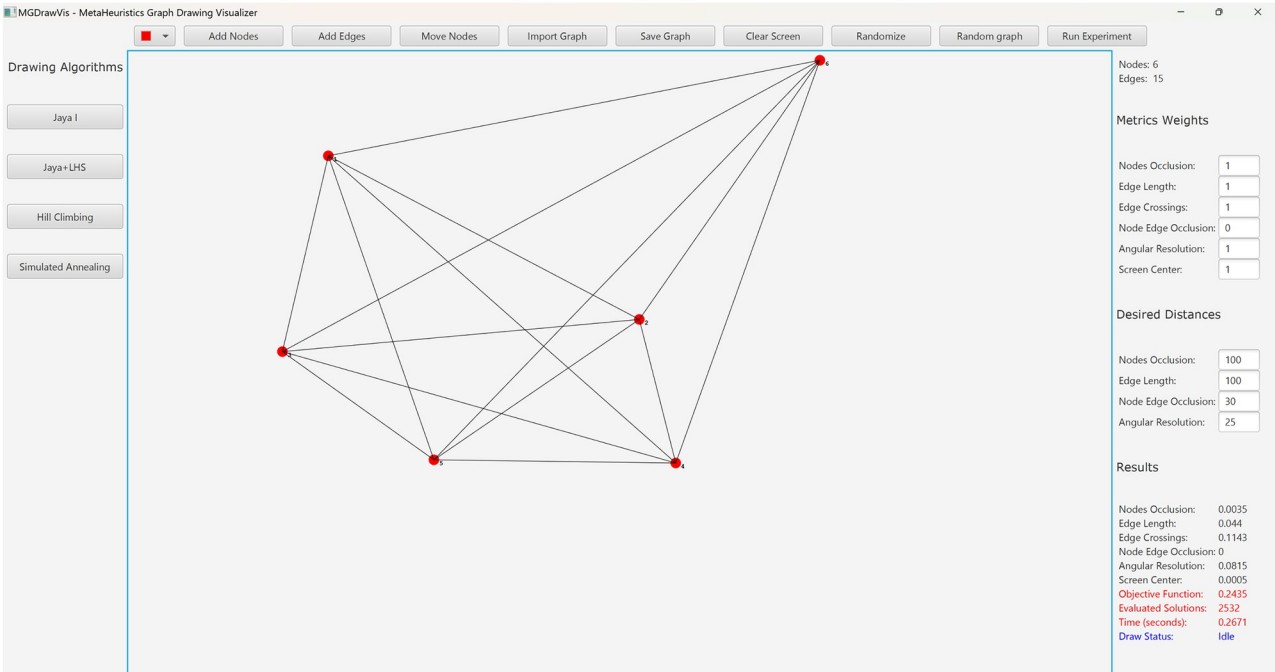

**Fig 1. Screenshot of our visualization tool MGDrawVis.**

wide range of functionalities make it an attractive option for researchers and practitioners in the field.

## 4.2. Tuning of methods

Metaheuristic tuning is a commonly practiced approach that involves finding good values for the algorithm's settings before running it, and then using these fixed values during the run [31].

For Hill Climbing and Jaya algorithm the parameter space is relatively small, so we could try all possible combinations of values and select the combination that resulted in the best objective function value. In contrast, for Simulated Annealing, we used an incremental tuning process, where we started with an initial value for each parameter and then tuned one parameter at a time while fixing the others. The reason for this approach is that the parameter space for Simulated Annealing is larger compared to Hill Climbing and Jaya algorithm, and trying all possible combinations of values would be computationally expensive. The specific values for each parameter were chosen based on the results of the tuning process and the performance of the optimization methods on the test cases. These tuning processes were previously used in [2, 6, 31].

To ensure that the selected parameter values were reliable, all methods were tested on 60 randomly generated graphs using Erdős-Rényi [32], where the locations of nodes and edges between the nodes were chosen randomly while ensuring that the graph is connected. Every edge had the same probability of appearing in the generated graph. For Hill Climbing, the method ran once on each test case as the method is deterministic, while for Simulated Annealing and Jaya algorithm, the methods ran 10 times on each test case, and the median was taken to avoid outliers as both methods are stochastic and include randomness. All methods started

**Table 1. Description of dataset used for parameters' tuning.**

| Group | Nodes | Edges | Test Cases |
|---|---|---|---|
| N50E100 | 50 | 100 | 20 |
| N100E200 | 100 | 200 | 20 |
| N150E300 | 150 | 300 | 20 |

from the same initial graph layout to ensure that the methods ran on the exact same datasets. The graphs' specifications are described in Table 1.

For Hill Climbing, we started the tuning process from the values that were found in previous work [19]. We tuned two parameters: the square size for the neighborhood search space and the reduction rate of the square when no better solution is found nearby. We tested different values for these parameters, including square sizes of 256, 512, 1024, and 2048, and reduction rates of 2, 4, and 6. See the heatmap in Fig 2.

For Jaya algorithm, we also used a combination of values for the two parameters: population size and the maximum number of iterations. We tested different values for these parameters, including population sizes of 10, 20, and 30, and maximum numbers of iterations of 20, 30, and 40. See the heatmap in Fig 3.

For Simulated Annealing, we also started the tuning process from the values that were selected in [19]. Then, we tested different values for the five parameters incrementally in the following order: square size, maximum number of iterations, number of iterations for each

| | | Objective Function | | |
|---|---|---|---|---|
| **SquareSize** | **Reduction** | **N50E100** | **N100E200** | **N150E300** |
| 256 | 2 | 3.8324 | 3.9304 | 3.9177 |
| 256 | 4 | 3.8521 | 3.9189 | 3.9216 |
| 256 | 6 | 3.8586 | 3.9196 | 3.9231 |
| 512 | 2 | 3.8101 | 3.8965 | 3.8616 |
| 512 | 4 | 3.7885 | 3.8592 | 3.8324 |
| 512 | 6 | 3.7729 | 3.8625 | 3.8183 |
| 1024 | 2 | 3.7887 | 3.9025 | 3.7931 |
| 1024 | 4 | 3.6808 | 3.8461 | 3.6348 |
| 1024 | 6 | 3.6801 | 3.6679 | 3.5874 |
| 2048 | 2 | 3.8243 | 3.8261 | 3.7022 |
| 2048 | 4 | 3.6878 | 3.6309 | 3.6878 |
| 2048 | 6 | 3.6748 | 3.7448 | 3.6584 |

**Fig 2. A heatmap of the examined values for Hill Climbing parameters.**

| Population Size | Iterations | Objective Function | | |
|---|---|---|---|---|
| | | N50E100 | N100E200 | N150E300 |
| 10 | 20 | 0.7334 | 1.3381 | 1.3569 |
| 10 | 30 | 0.7479 | 1.1784 | 1.2853 |
| 10 | 40 | 0.7832 | 1.0488 | 1.1868 |
| 20 | 20 | 0.9557 | 1.3042 | 1.4833 |
| 20 | 30 | 0.8285 | 1.2174 | 1.2825 |
| 20 | 40 | 0.8525 | 1.1466 | 1.2168 |
| 30 | 20 | 0.9982 | 1.4157 | 1.2674 |
| 30 | 30 | 0.8115 | 1.1901 | 1.4999 |
| 30 | 40 | 0.9323 | 1.2178 | 1.2939 |

**Fig 3. A heatmap of the examined values for Jaya algorithm parameters.**

temperature, initial temperature, and cooling down rate. We started with an initial square size of 256 and tested sizes of 512 and 1024. We tested maximum numbers of iterations of 20, 30, 40, 50, and 60, and numbers of iterations for each temperature of 10, 15, and 20. For the initial temperature, we tested values of 0.55, 0.65, 0.75, and 0.85, and for the cooling down rate, we tested values of 0.6, 0.7, 0.8, and 0.9. See the heatmap in Fig 4.

Based on the test results described in the above tables, we selected the following parameter values for each method:

- Hill Climbing: Square size = 1024, Reduction = 6;

- Simulated Annealing: Square size = 512, Maximum number of iterations = 50, Iterations per temperature = 15, Initial temperature = 0.65, Cooling down rate = 0.7;

- Jaya: Population size = 10, Maximum number of iterations = 40

In the following sections we explore the performance of these methods using the selected parameter values, to determine the effectiveness and efficiency of each method for graph layout optimization.

### 4.3. Performance comparison on random graphs

This section starts our investigation into our two main research questions: Firstly, does Jaya algorithm perform better than Hill Climbing and Simulated Annealing approaches in the field of graph layout? Secondly, does applying Latin Hypercube Sampling (LHS) for initializing the population for Jaya algorithm improve the performance of the original Jaya graph drawing algorithm? We formulated four specific research objectives to guide our investigation. These are:

1. What is the highest quality layout achievable with Jaya algorithm compared to the other algorithms, as measured by a weighted sum objective function?

| SquareSize | Max Itr. | Inner Itr. | Temp. | Cool. Dn. | Objective Function | | |
|---|---|---|---|---|---|---|---|
| | | | | | N50E100 | N100E200 | N150E300 |
| 256 | 40 | 15 | 0.75 | 0.80 | 1.9851 | 2.1520 | 2.0968 |
| 512 | 40 | 15 | 0.75 | 0.80 | 1.8070 | 2.1778 | 2.0592 |
| 1024 | 40 | 15 | 0.75 | 0.80 | 1.7823 | 2.5380 | 2.8465 |
| 512 | 20 | 15 | 0.75 | 0.80 | 2.4293 | 3.3749 | 3.2728 |
| 512 | 30 | 15 | 0.75 | 0.80 | 1.7604 | 2.5528 | 2.2208 |
| 512 | 40 | 15 | 0.75 | 0.80 | 1.8070 | 2.1778 | 2.0592 |
| 512 | 50 | 15 | 0.75 | 0.80 | 1.6602 | 2.4449 | 1.9009 |
| 512 | 60 | 15 | 0.75 | 0.80 | 1.6711 | 2.3291 | 1.9010 |
| 512 | 50 | 10 | 0.75 | 0.80 | 1.4730 | 2.3233 | 2.1028 |
| 512 | 50 | 15 | 0.75 | 0.80 | 1.6602 | 2.4449 | 1.9009 |
| 512 | 50 | 20 | 0.75 | 0.80 | 1.7653 | 2.3180 | 2.1811 |
| 512 | 50 | 15 | 0.55 | 0.80 | 1.5579 | 2.8497 | 1.9278 |
| 512 | 50 | 15 | 0.65 | 0.80 | 1.5119 | 2.1400 | 1.9392 |
| 512 | 50 | 15 | 0.75 | 0.80 | 1.6602 | 2.4449 | 1.9009 |
| 512 | 50 | 15 | 0.85 | 0.80 | 1.7725 | 2.1728 | 1.9983 |
| 512 | 50 | 15 | 0.65 | 0.60 | 1.5313 | 2.1841 | 2.3013 |
| 512 | 50 | 15 | 0.65 | 0.70 | 1.5326 | 1.9986 | 1.8492 |
| 512 | 50 | 15 | 0.65 | 0.90 | 2.0910 | 2.8933 | 2.8951 |

**Fig 4. A heatmap of the examined values for simulated annealing parameters.**

2. How the Jaya algorithm compare to the other algorithms in terms of producing high-quality graph layouts within a limited number of computational resources?

3. How quickly can Jaya algorithm produce high quality layouts compared to the other algorithms?

4. How does the use of Latin Hypercube initialization improve the performance of the Jaya algorithm in graph layout?

To address these objectives, we conducted a series of experiments on randomly generated and real-world datasets with graphs of varying sizes.

To assess the performance of Jaya algorithm and its Latin Hypercube Sampling (LHS) version compared to Hill Climbing and Simulated Annealing, we generated random graph datasets with varying numbers of nodes and edges using the Erdős–Rényi model. The random graph generator used the number of nodes and edges as parameters and randomly selected locations for the nodes based on the size of the graph display window. The generator also randomly chose endpoints for edges while ensuring that self-sourcing edges and multiple edges between the same pair of nodes were not allowed. Moreover, only connected graphs were accepted.

We chose to use the Erdős–Rényi model because it is a widely studied and commonly used model for generating random graphs that is both simple and computationally efficient, making it an ideal choice for generating large and diverse datasets for experimental studies.

To compare the methods, we first applied our Jaya-based approach, along with Hill Climbing and Simulated Annealing, to the generated graphs. While Hill Climbing is a deterministic method that is not affected by randomness, Jaya algorithm and Simulated Annealing are stochastic methods that use random elements in the neighborhood searching process. Therefore, we tested each method on each individual graph for 10 different runs and computed the median of the results to compare the outcomes.

To make a comprehensive comparison, we divided our experiment into three phases. In order to answer Research Objective (4), we tested both Jaya algorithm with a standard random initial layout and Jaya algorithm with Latin Hypercube Sampling initialization throughout the following phases.

To ensure the robustness and generalizability of our results, we used different datasets for each phase of our experiment. Using the same dataset for all phases can result in overfitting, where the algorithms are optimized to perform well on that specific dataset but may not perform as well on new or different datasets. By using different datasets for each phase, we aimed to test the algorithms' performance on a wider range of graph layouts, which can provide more robust and generalizable results. Additionally, using different datasets helped to ensure that the results were not biased towards any specific characteristics of the dataset used.

Phase I was designed to answer Research Objective (1). Here, we applied the methods to the first dataset described in Table 2. The dataset contained four graph groups, each consisting of 10 test cases with the same number of nodes and edges but different graphs and initial layouts. The first group had 50 nodes and 100 edges, with the number of nodes increasing by 50 for each subsequent graph group, and the number of edges increasing by 100, ending with graphs having 200 nodes and 400 edges.

For each group, we executed the methods on 10 test cases and computed the average final objective function value and the average number of evaluated solutions required to reach the final value for each method. In this phase, we ran the methods with the previously discussed parameters.

The results of phase I are presented in three bar charts. Fig 5 and Table 3 show the difference between the methods in terms of the lowest value of the objective function value that can be achieved. Fig 6 and Table 4 show the number of evaluated solutions required to achieve this objective function value, and Fig 7 and Table 5 show the execution time in seconds for each method.

Phase II was designed to answer Research Objective (2). Here, we aimed to evaluate the quality of the layout produced by the drawing algorithms within a fixed number of computational resources. To achieve this, we generated another dataset that contained five graph groups, each consisting of 10 test cases with the same number of nodes and edges but different graphs and initial layouts, as described in Table 6.

We used the same number of evaluated solutions for each method. First, we ran the Hill Climbing method on the graphs and recorded the number of evaluated solutions required to reach the lowest objective function value. We used Hill Climbing as a baseline because,

**Table 2. Description of dataset used in Phase I experiment.**

| Group | Nodes | Edges | Test Cases |
|---|---|---|---|
| N50E100 | 50 | 100 | 10 |
| N100E200 | 100 | 200 | 10 |
| N150E300 | 150 | 300 | 10 |
| N200E400 | 200 | 400 | 10 |

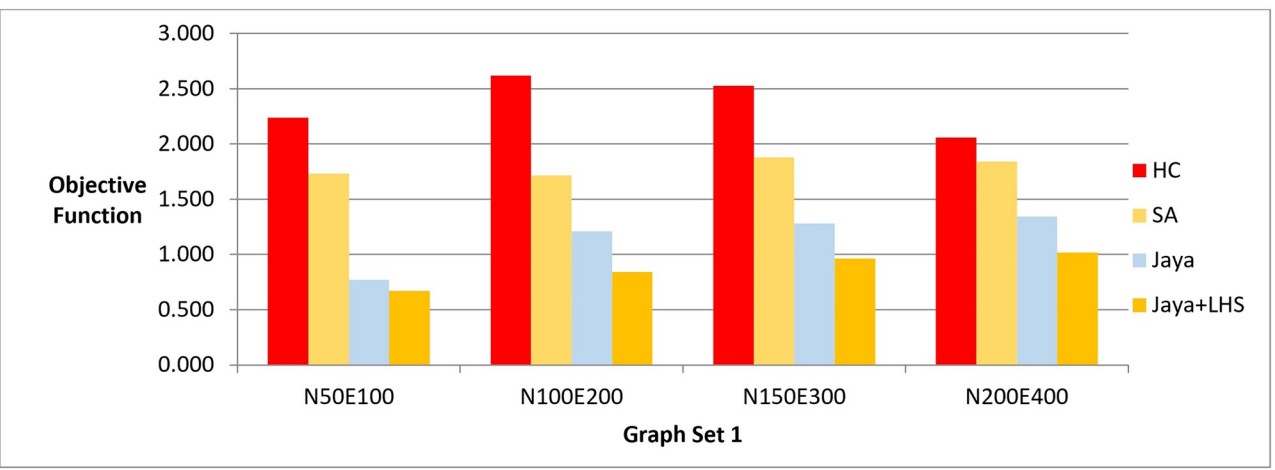

**Fig 5. A bar chart for the values of objective function—Phase I.**

**Table 3. Summary statistics for the values of objective function—Phase I.**

| | Objective Function | | | | | | | | | | | | | | |
|---|---|---|---|---|---|---|---|---|---|---|---|---|---|---|---|
| | Hill Climbing | | | | Simulated Annealing | | | | Jaya | | | | Jaya + LHS | | | |
| Graph Set | Mean | Median | Max | Min | Mean | Median | Max | Min | Mean | Median | Max | Min | Mean | Median | Max | Min |
| N50E100 | 2.240 | 2.380 | 2.520 | 2.040 | 1.732 | 1.670 | 2.118 | 1.419 | 0.770 | 0.776 | 0.910 | 0.636 | 0.667 | 0.655 | 0.738 | 0.614 |
| N100E200 | 2.617 | 2.603 | 2.918 | 2.315 | 1.716 | 1.748 | 1.839 | 1.485 | 1.211 | 1.219 | 1.377 | 0.995 | 0.841 | 0.805 | 0.948 | 0.741 |
| N150E300 | 2.527 | 2.546 | 2.813 | 2.267 | 1.878 | 1.961 | 2.088 | 1.644 | 1.281 | 1.274 | 1.442 | 1.095 | 0.961 | 0.996 | 1.068 | 0.796 |
| N200E400 | 2.058 | 2.105 | 2.371 | 1.809 | 1.839 | 1.836 | 1.994 | 1.740 | 1.342 | 1.336 | 1.487 | 1.206 | 1.017 | 1.059 | 1.074 | 0.881 |

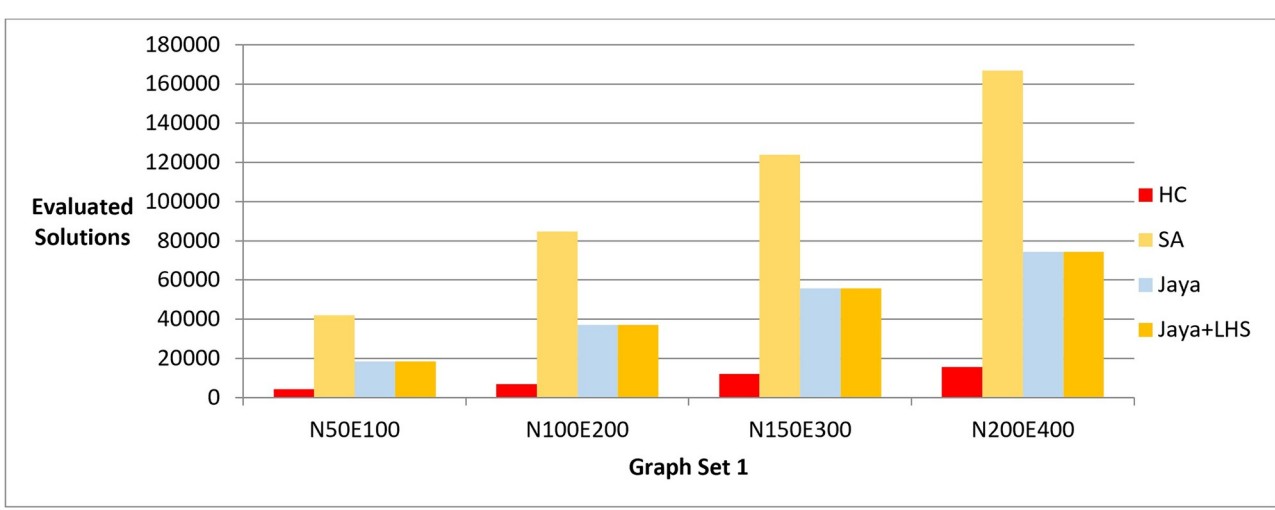

**Fig 6. A bar chart for the number of function evaluations—Phase I.**

**Table 4. Summary statistics for the number of function evaluations—Phase I.**

| | Evaluated Solutions | | | | | | | | | | | | | | | |
| --- | --- | --- | --- | --- | --- | --- | --- | --- | --- | --- | --- | --- | --- | --- | --- | --- |
| | Hill Climbing | | | | Simulated Annealing | | | | Jaya | | | | Jaya + LHS | | | |
| Graph Set | Mean | Median | Max | Min | Mean | Median | Max | Min | Mean | Median | Max | Min | Mean | Median | Max | Min |
| N50E100 | 4430 | 4487 | 4642 | 4268 | 41998 | 41960 | 42485 | 41675 | 18600 | 18600 | 18600 | 18600 | 18600 | 18600 | 18600 | 18600 |
| N100E200 | 6988 | 6912 | 7066 | 6811 | 84804 | 84680 | 85610 | 84115 | 37200 | 37200 | 37200 | 37200 | 37200 | 37200 | 37200 | 37200 |
| N150E300 | 12228 | 12327 | 12531 | 12019 | 123893 | 124185 | 125325 | 122065 | 55800 | 55800 | 55800 | 55800 | 55800 | 55800 | 55800 | 55800 |
| N200E400 | 15550 | 15550 | 15622 | 15401 | 166871 | 166625 | 168435 | 165815 | 74400 | 74400 | 74400 | 74400 | 74400 | 74400 | 74400 | 74400 |

according to the results of phase I, it generated the graph layout with the least number of evaluated solutions, which means all other methods can reach that number.

Next, we ran Jaya algorithm and Simulated Annealing methods for each graph layout until they performed the same number of evaluated solutions as the baseline Hill Climbing method. Finally, we measured the value of the objective function produced by the drawing algorithms. This gave us the objective function when the methods applied the same number of evaluated solutions. The results obtained from phase II are demonstrated in Fig 8 and Table 7.

Phase III was designed to answer Research Objective (3). Here, we investigated the performance of the approaches rather than the quality of the produced layouts. For this phase, we generated another dataset that contained five graph groups, each consisting of 10 test cases with the same number of nodes and edges but different graphs and initial layouts, as described in Table 8.

To test which method has the fewest number of evaluated solutions to reach similar values for the objective function, we first ran the Hill Climbing method on the graphs until no further improvements could be made on the value of the objective function. As with Phase II, we used Hill Climbing as a baseline because, in phase I, it produced graph layouts with highest value of the objective function compared to the other methods. Therefore, we assumed that Jaya algorithm and Simulated Annealing could typically reach the value of the objective function produced by Hill Climbing. We then ran Jaya algorithm and Simulated Annealing methods until

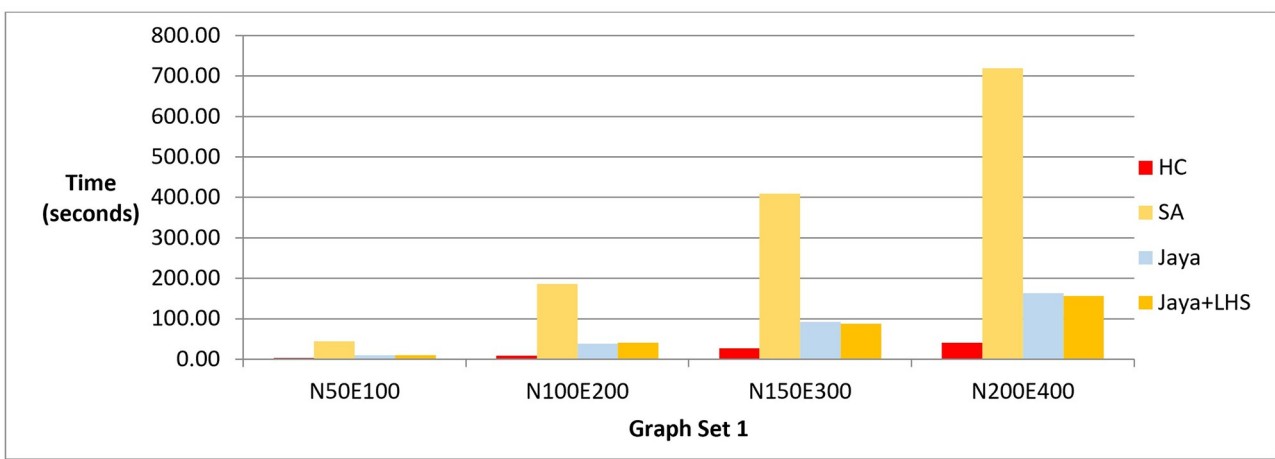

**Fig 7. A bar chart for the execution time in seconds—Phase I.**

**Table 5. Summary statistics for the execution time in seconds—Phase I.**

| | Time (seconds) | | | | | | | | | | | | | | | |
|---|---|---|---|---|---|---|---|---|---|---|---|---|---|---|---|---|
| | Hill Climbing | | | | Simulated Annealing | | | | Jaya | | | | Jaya + LHS | | | |
| Graph Set | Mean | Median | Max | Min | Mean | Median | Max | Min | Mean | Median | Max | Min | Mean | Median | Max | Min |
| N50E100 | 2.83 | 2.87 | 3.01 | 2.56 | 43.89 | 43.86 | 46.02 | 42.17 | 9.52 | 9.46 | 9.77 | 9.38 | 9.11 | 9.02 | 9.48 | 8.96 |
| N100E200 | 8.74 | 8.84 | 9.71 | 8.01 | 185.81 | 185.25 | 190.42 | 183.13 | 38.47 | 38.33 | 40.05 | 37.75 | 40.98 | 40.73 | 41.75 | 40.16 |
| N150E300 | 26.45 | 26.65 | 27.25 | 26.01 | 408.95 | 409.72 | 419.80 | 402.01 | 91.99 | 92.67 | 95.05 | 89.27 | 87.27 | 84.47 | 95.40 | 83.93 |
| N200E400 | 41.00 | 41.01 | 41.90 | 40.00 | 719.62 | 719.63 | 744.95 | 703.75 | 163.21 | 163.22 | 166.92 | 158.66 | 155.56 | 155.28 | 162.09 | 147.72 |

they reached the objective function value found by Hill Climbing and measured the number of evaluated solutions for each method. The results obtained from phase III are presented in Fig 9 and Table 9.

Fig 10 shows the layouts of a random graph, with 6 nodes and 15 edges, drawn by Hill climbing, Simulated Annealing, Jaya algorithm, and Jaya algorithm with Latin Hypercube Sampling initialization.

In this study, we examined one deterministic and three stochastic algorithms for graph layout optimization. The deterministic algorithm was applied once on the same initial graph layout, while the stochastic methods were applied 10 times on the same graph to account for randomness. The main internal threat to validity is related to the implementation of the algorithms, as all methods were implemented by the same programmer and run on the same machine. There is a possibility that one of the methods was implemented in a more efficient way, but they share a substantial code and calculation of the objective function, which increases confidence that none was particularly disadvantaged. Additionally, we applied a systematic parameter tuning method to ensure fair comparison.

As for external threats to validity, there is a risk that the results may not be generalizable to real-world graphs. However, we aimed to reduce selection bias by using randomly generated graphs (with the exception of parameters such as the number of nodes and edges in the generation algorithm). Despite this, randomly generated graphs may not have the same characteristics as real-world graphs. To address this, we explore the methods applied to real-world standard public datasets in the next section.

## 4.4. Performance comparison on real-world graphs

Following the conduction of multiple experiments on random graphs, we evaluated the performance of our proposed Jaya methods against Hill Climbing and Simulated Annealing on real-world graph datasets to examine if they produced comparable results in a practical setting. We chose 10 distinct datasets with varying graph sizes and features from different sources, as presented in Table 10, which also includes information on the number of nodes, edges, and

**Table 6. Description of dataset used in Phase II experiment.**

| Group | Nodes | Edges | Test Cases |
|---|---|---|---|
| N30E63 | 30 | 63 | 10 |
| N47E81 | 47 | 81 | 10 |
| N90E130 | 90 | 130 | 10 |
| N140E200 | 140 | 200 | 10 |
| N175E300 | 175 | 300 | 10 |

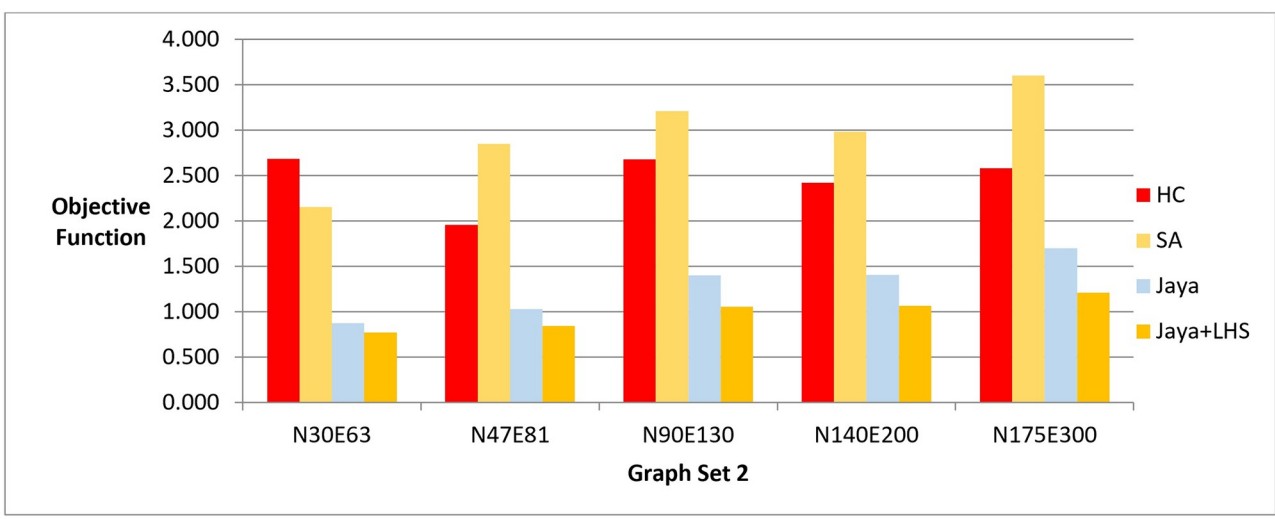

**Fig 8. A bar chart for the values of objective function—Phase II.**

description for each dataset. The initial positions of the nodes in each graph were randomly generated. The Hill Climbing method was executed once for each dataset, whereas the Simulated Annealing and both variants of Jaya algorithm were executed 10 times, following the same procedure as in the previous experiments. For each 10 runs, the median value was determined and used for comparison with the results obtained by the other methods. We evaluated the performance of the methods using phases I, II, and III, which are given in the previous section. The outcomes of the experiments are presented in the following figures and tables, where Fig 11, Table 11, Fig 12, and Table 12 display the results of applying the methods on real data graphs as described in Table 10 according to phase I.

In reference to phase II, Fig 13 and Table 13 illustrate the objective function values obtained by each method during testing on the real-world graphs listed in Table 10. Fig 14 and Table 14, on the other hand, show the number of evaluated solutions produced by each method when the experiment outlined in phase III was implemented on the same dataset.

Fig 15 shows the visual representation of the layouts generated by all the methods applied to graph 3, which is "03_moreno_zebra" in Table 10. This graph contains 28 nodes and 118 edges.

**Table 7. Summary statistics for the values of objective function—Phase II.**

| | Objective Function | | | | | | | | | | | | | | | |
|---|---|---|---|---|---|---|---|---|---|---|---|---|---|---|---|---|
| | Hill Climbing | | | | Simulated Annealing | | | | Jaya | | | | Jaya + LHS | | | |
| Graph Set | Mean | Median | Max | Min | Mean | Median | Max | Min | Mean | Median | Max | Min | Mean | Median | Max | Min |
| N30E63 | 2.683 | 2.591 | 2.921 | 2.321 | 2.151 | 2.198 | 2.461 | 1.667 | 0.876 | 0.973 | 1.070 | 0.605 | 0.771 | 0.760 | 0.889 | 0.595 |
| N47E81 | 1.958 | 1.902 | 2.413 | 1.625 | 2.846 | 2.944 | 3.121 | 2.524 | 1.030 | 1.043 | 1.125 | 0.931 | 0.846 | 0.895 | 0.975 | 0.549 |
| N90E130 | 2.675 | 2.632 | 2.885 | 2.491 | 3.207 | 3.241 | 3.410 | 2.946 | 1.398 | 1.403 | 1.583 | 1.248 | 1.052 | 1.110 | 1.150 | 0.894 |
| N140E200 | 2.422 | 2.449 | 2.712 | 2.223 | 2.983 | 2.980 | 3.204 | 2.776 | 1.404 | 1.406 | 1.499 | 1.267 | 1.067 | 1.029 | 1.206 | 0.924 |
| N175E300 | 2.580 | 2.561 | 2.782 | 2.319 | 3.598 | 3.622 | 3.699 | 3.457 | 1.698 | 1.729 | 1.759 | 1.519 | 1.211 | 1.253 | 1.296 | 0.995 |

**Table 8. Description of dataset used in Phase III experiment.**

| Group | Nodes | Edges | Test Cases |
|---|---|---|---|
| N40E70 | 40 | 70 | 10 |
| N65E95 | 65 | 95 | 10 |
| N100E145 | 100 | 145 | 10 |
| N160E210 | 160 | 210 | 10 |
| N210E350 | 210 | 350 | 10 |

## 4.5. Scalability

To investigate the scalability of the drawing algorithms, we used a randomly generated dataset consisting of graphs ranging from 50 to 500 nodes and from 100 to 1000 edges, with the number of nodes increasing by 50 and the number of edges increasing by 100 for each graph as described in Table 15. While we would have preferred to include larger graphs in our analysis, we found it necessary to limit the maximum graph size to 500 nodes due to computational limitations with the Simulated Annealing algorithm. Beyond this size, the algorithm took a long time to execute and was four times slower than Jaya algorithm, which led us to conclude that it was no longer practical to continue testing with larger graphs.

We executed each algorithm on the graph dataset, measuring the quality of the produced layouts, number of evaluated solutions, and the computation time required to produce them. Figs 16–18 show the overall performance of our method when being applied on a set of graphs with increasing number of nodes and edges, as described in the table above, in terms of objective function's values, number of evaluated solutions, and execution time in seconds respectively.

It is worth noting that our decision to limit the maximum graph size to 500 nodes was based on practical considerations related to the available computational resources and the execution time of the algorithms. While it is possible that our results could have differed if we had been able to include larger graphs, we believe that our analysis provides useful insights into the

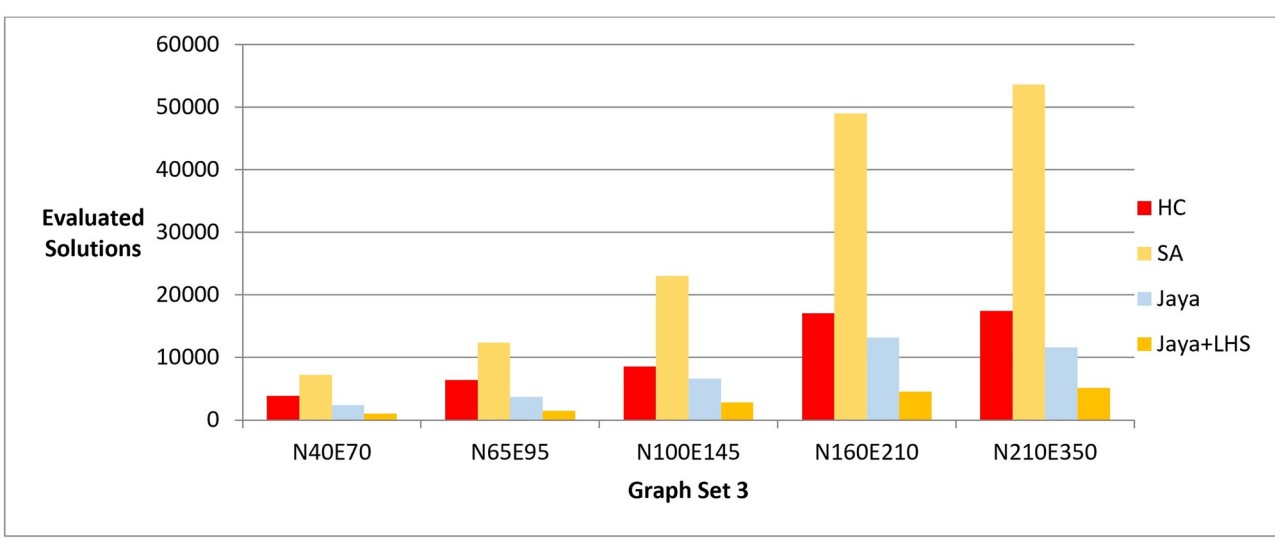

**Fig 9. A bar chart for the number of function evaluations—Phase III.**

**Table 9. Summary statistics for the number of function evaluations—Phase III.**

| Graph Set | Hill Climbing | | | | Simulated Annealing | | | | Jaya | | | | Jaya + LHS | | | |
|---|---|---|---|---|---|---|---|---|---|---|---|---|---|---|---|---|
| | Mean | Median | Max | Min | Mean | Median | Max | Min | Mean | Median | Max | Min | Mean | Median | Max | Min |
| N40E70 | 3835 | 3867 | 4012 | 3677 | 7201 | 6865 | 8945 | 6160 | 2352 | 1920 | 3720 | 1560 | 1056 | 1200 | 1200 | 840 |
| N65E95 | 6390 | 6321 | 6541 | 6190 | 12397 | 12040 | 13205 | 11560 | 3705 | 3705 | 4875 | 1950 | 1482 | 1365 | 1950 | 780 |
| N100E145 | 8545 | 8499 | 8719 | 8266 | 23003 | 21305 | 28505 | 20680 | 6600 | 6600 | 8400 | 4800 | 2820 | 3000 | 3000 | 2100 |
| N160E210 | 17098 | 17003 | 17623 | 16721 | 49008 | 49715 | 56695 | 37450 | 13152 | 12000 | 17760 | 10560 | 4512 | 4800 | 4800 | 3360 |
| N210E350 | 17455 | 17534 | 17821 | 17184 | 53646 | 52480 | 61585 | 47575 | 11592 | 8190 | 21420 | 4410 | 5166 | 4410 | 6300 | 4410 |

scalability of these drawing algorithms in the context of the graph sizes commonly encountered in practice.

## 4.6. Performance

We aimed to investigate the performance of our method concerning the objective function's value as the count of evaluated solutions rises. To achieve this, we applied our method to

Initial layout for graph with 6 nodes and 15 edges

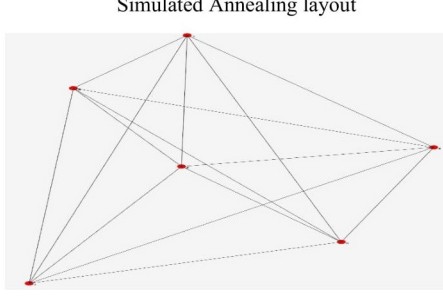

Hill Climbing layout
Simulated Annealing layout

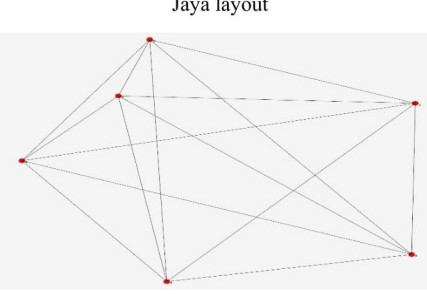

Jaya layout
Jaya with LHS layout

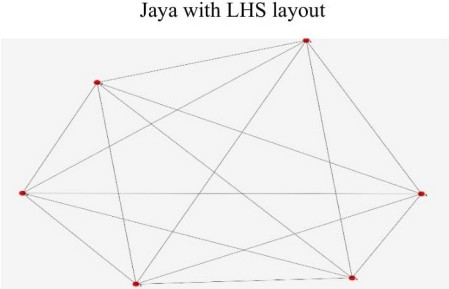

**Fig 10. Layouts of a random graph with 6 nodes and 15 edges.**

**Table 10. Description of the real-world dataset.**

| Graph | Nodes | Edges | Description | Source |
|---|---|---|---|---|
| 01_dolphins | 62 | 159 | This dataset features an undirected social network representing frequent interactions among 62 dolphins inhabiting Doubtful Sound, New Zealand. | [33] |
| 02_contiguous_usa | 49 | 111 | This dataset represents the contiguous United States and the District of Columbia, consisting of 48 states and DC, excluding Alaska and Hawaii, with edges denoting shared borders between states. | [34] |
| 03_moreno_zebra | 28 | 118 | The undirected network depicts interactions among 28 Grévy's zebras in Kenya, with nodes representing individual zebras and edges signifying observed interactions during the study. | [35] |
| 04_simulated_BlockModelGraph_50Nodes | 50 | 319 | This dataset originates from the 2017 Streaming Partition Challenge, containing data streams with known truth partitions for evaluating community detection algorithms. | [36] |
| 05_simulated_BlockModelGraph_100Nodes | 100 | 778 | This dataset originates from the 2017 Streaming Partition Challenge, containing data streams with known truth partitions for evaluating community detection algorithms. | [36] |
| 06_football | 115 | 613 | This dataset features the network of Division IA college American football games that took place during the Fall 2000 regular season. | [37] |
| 07_karate | 34 | 78 | This dataset presents the network of friendships among 34 members of a karate club at a US university. | [38] |
| 08_lesmis | 77 | 254 | This dataset contains a weighted network of character co-appearances in Victor Hugo's "Les Misérables," with nodes representing characters and edges connecting those who appear together in a chapter. | [39] |
| 09_celegansneural | 297 | 2148 | This dataset represents a weighted, directed network depicting the neural network of C. Elegans. | [40] |
| 10_USAirLines | 332 | 2126 | This dataset consists of undirected weighted graphs representing US air flights between various airports. | [41] |

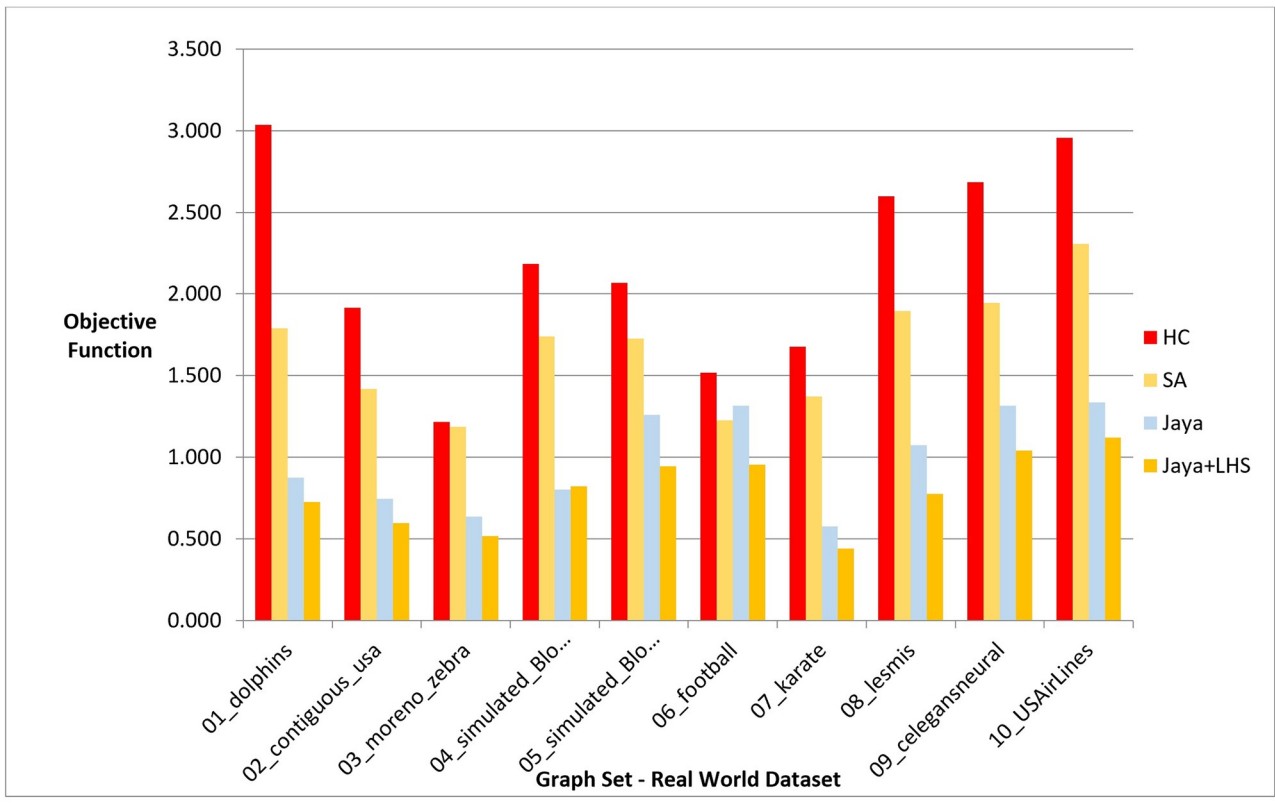

**Fig 11. A bar chart for the values of objective function on real-world dataset—Phase I.**

**Table 11. Summary statistics for the values of objective function on real-world dataset—Phase I.**

| Graph Set | Objective Function | | | | | | | | | | | | | | | |
|---|---|---|---|---|---|---|---|---|---|---|---|---|---|---|---|---|
| | Hill Climbing | | | | Simulated Annealing | | | | Jaya | | | | Jaya + LHS | | | |
| | Mean | Median | Max | Min | Mean | Median | Max | Min | Mean | Median | Max | Min | Mean | Median | Max | Min |
| 01_dolphins | 3.036 | 3.036 | 3.036 | 3.036 | 1.790 | 1.916 | 2.067 | 1.386 | 0.873 | 0.886 | 0.960 | 0.772 | 0.726 | 0.738 | 0.831 | 0.611 |
| 02_contiguous_usa | 1.915 | 1.915 | 1.915 | 1.915 | 1.419 | 1.396 | 1.571 | 1.330 | 0.746 | 0.713 | 0.864 | 0.663 | 0.595 | 0.589 | 0.670 | 0.527 |
| 03_moreno_zebra | 1.214 | 1.214 | 1.214 | 1.214 | 1.186 | 1.226 | 1.304 | 1.028 | 0.636 | 0.661 | 0.800 | 0.448 | 0.518 | 0.539 | 0.578 | 0.437 |
| 04_simulated_BlockModelGraph_50Nodes | 2.183 | 2.183 | 2.183 | 2.183 | 1.738 | 1.721 | 1.822 | 1.673 | 0.803 | 0.827 | 0.863 | 0.718 | 0.822 | 0.873 | 0.926 | 0.665 |
| 05_simulated_BlockModelGraph_100Nodes | 2.069 | 2.069 | 2.069 | 2.069 | 1.726 | 1.742 | 1.877 | 1.558 | 1.259 | 1.320 | 1.382 | 1.073 | 0.943 | 0.979 | 1.042 | 0.807 |
| 06_football | 1.517 | 1.517 | 1.517 | 1.517 | 1.226 | 1.226 | 1.301 | 1.151 | 1.315 | 1.369 | 1.386 | 1.191 | 0.954 | 1.000 | 1.068 | 0.795 |
| 07_karate | 1.678 | 1.678 | 1.678 | 1.678 | 1.373 | 1.331 | 1.577 | 1.210 | 0.575 | 0.629 | 0.714 | 0.382 | 0.440 | 0.457 | 0.486 | 0.377 |
| 08_lesmis | 2.597 | 2.597 | 2.597 | 2.597 | 1.896 | 2.058 | 2.112 | 1.517 | 1.072 | 1.111 | 1.205 | 0.900 | 0.774 | 0.837 | 0.844 | 0.640 |
| 09_celegansneural | 2.685 | 2.685 | 2.685 | 2.685 | 1.946 | 1.901 | 2.121 | 1.802 | 1.314 | 1.312 | 1.336 | 1.295 | 1.040 | 1.119 | 1.161 | 0.840 |
| 10_USAirLines | 2.955 | 2.955 | 2.955 | 2.955 | 2.308 | 2.311 | 2.512 | 2.231 | 1.337 | 1.389 | 1.457 | 1.164 | 1.120 | 1.180 | 1.233 | 0.946 |

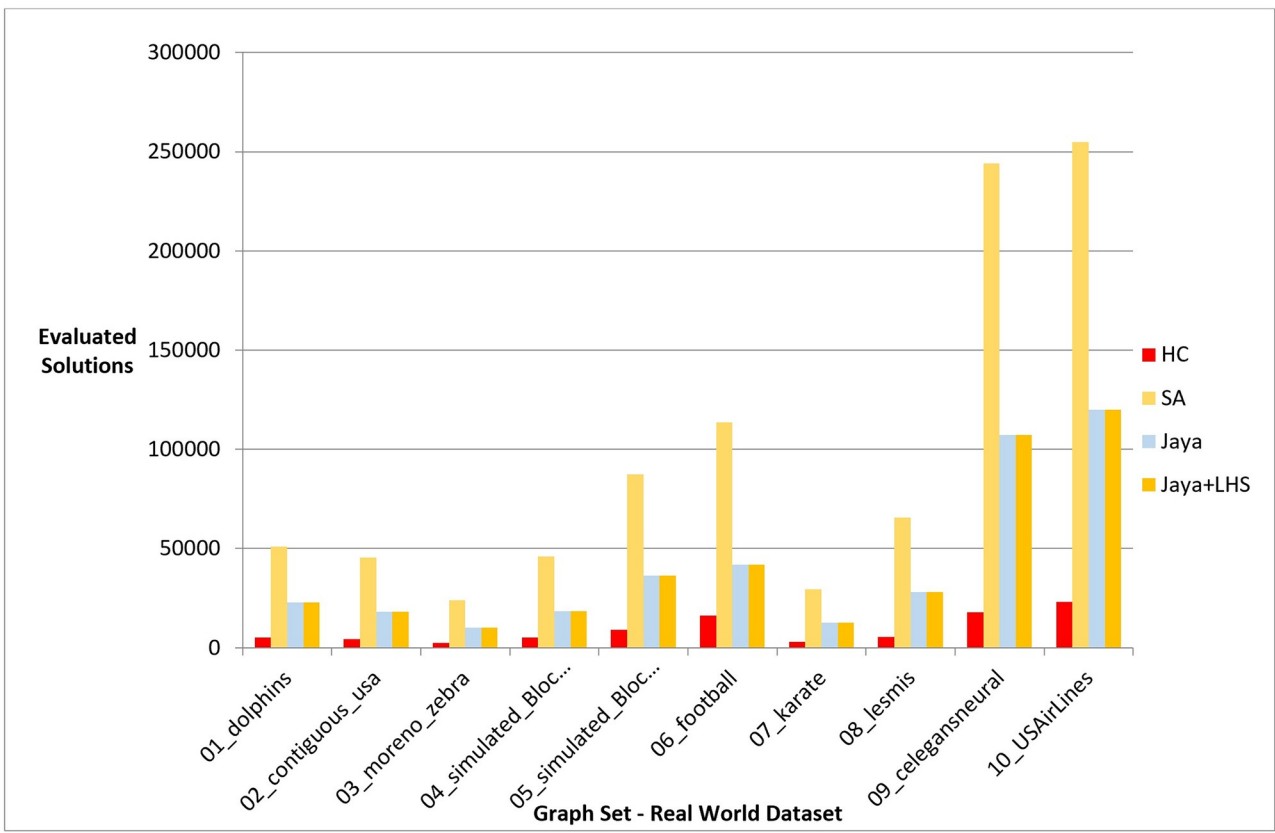

**Fig 12. A bar chart for the number of function evaluations on real-world dataset—Phase I.**

multiple graphs, all maintaining the same size—consisting of 100 nodes and 150 edges—but with varied initial layouts. We tracked and recorded the average value of the objective function at several intervals throughout the method's execution time. Fig 19 presents the fluctuation in the objective function's value corresponding to the increasing number of evaluated solutions.

## 5. Analysis of results

The objective of this study was to evaluate our proposed methods with different graph drawing methods and their capabilities in terms of producing high-quality graph layouts with minimum evaluated solutions. In order to achieve this, we tested Hill Climbing, Simulated Annealing, and the two proposed versions of Jaya algorithm with Latin Hypercube Sampling (LHS) on random and real-world graph datasets. The evaluation was conducted based on three different phases, namely phase I, phase II, and phase III as described in the previous section.

In phase I, we aimed to find the best layout with the lowest objective function value. The results showed that Jaya algorithm with LHS produced the best graph layouts compared to the original Jaya algorithm, Simulated Annealing, and Hill Climbing as shown in Fig 5. Simulated Annealing evaluated the most solutions to achieve the best layout, followed by both versions of Jaya algorithm and Hill Climbing as shown in Fig 6. This was also confirmed by the execution time of each method demonstrated in Fig 7. To verify the statistical significance of the results,

**Table 12. Summary statistics for the number of function evaluations on real-world dataset—Phase I.**

| | Evaluated Solutions | | | | | | | | | | | | | | | |
|---|---|---|---|---|---|---|---|---|---|---|---|---|---|---|---|---|
| | Hill Climbing | | | | Simulated Annealing | | | | Jaya | | | | Jaya + LHS | | | |
| Graph Set | Mean | Median | Max | Min | Mean | Median | Max | Min | Mean | Median | Max | Min | Mean | Median | Max | Min |
| 01_dolphins | 5155 | 5155 | 5155 | 5155 | 51009 | 51066 | 52208 | 50353 | 22692 | 22692 | 22692 | 22692 | 22692 | 22692 | 22692 | 22692 |
| 02_contiguous_usa | 4161 | 4161 | 4161 | 4161 | 45291 | 45461 | 45963 | 44450 | 18012 | 18012 | 18012 | 18012 | 18012 | 18012 | 18012 | 18012 |
| 03_moreno_zebra | 2292 | 2292 | 2292 | 2292 | 23744 | 23949 | 24020 | 23263 | 10092 | 10092 | 10092 | 10092 | 10092 | 10092 | 10092 | 10092 |
| 04_simulated_BlockModelGraph_50Nodes | 4987 | 4987 | 4987 | 4987 | 46021 | 46375 | 46676 | 45011 | 18372 | 18372 | 18372 | 18372 | 18372 | 18372 | 18372 | 18372 |
| 05_simulated_BlockModelGraph_100Nodes | 9026 | 9026 | 9026 | 9026 | 87478 | 87124 | 89158 | 86151 | 36372 | 36372 | 36372 | 36372 | 36372 | 36372 | 36372 | 36372 |
| 06_football | 16108 | 16108 | 16108 | 16108 | 113624 | 112712 | 115486 | 112675 | 41772 | 41772 | 41772 | 41772 | 41772 | 41772 | 41772 | 41772 |
| 07_karate | 2958 | 2958 | 2958 | 2958 | 29411 | 29325 | 30075 | 28832 | 12612 | 12612 | 12612 | 12612 | 12612 | 12612 | 12612 | 12612 |
| 08_lesmis | 5342 | 5342 | 5342 | 5342 | 65572 | 65997 | 66794 | 63925 | 28092 | 28092 | 28092 | 28092 | 28092 | 28092 | 28092 | 28092 |
| 09_celegansneural | 17804 | 17804 | 17804 | 17804 | 243949 | 243121 | 244110 | 242102 | 107292 | 107292 | 107292 | 107292 | 107292 | 107292 | 107292 | 107292 |
| 10_USAirLines | 22974 | 22974 | 22974 | 22974 | 254820 | 253991 | 254981 | 252019 | 119892 | 119892 | 119892 | 119892 | 119892 | 119892 | 119892 | 119892 |

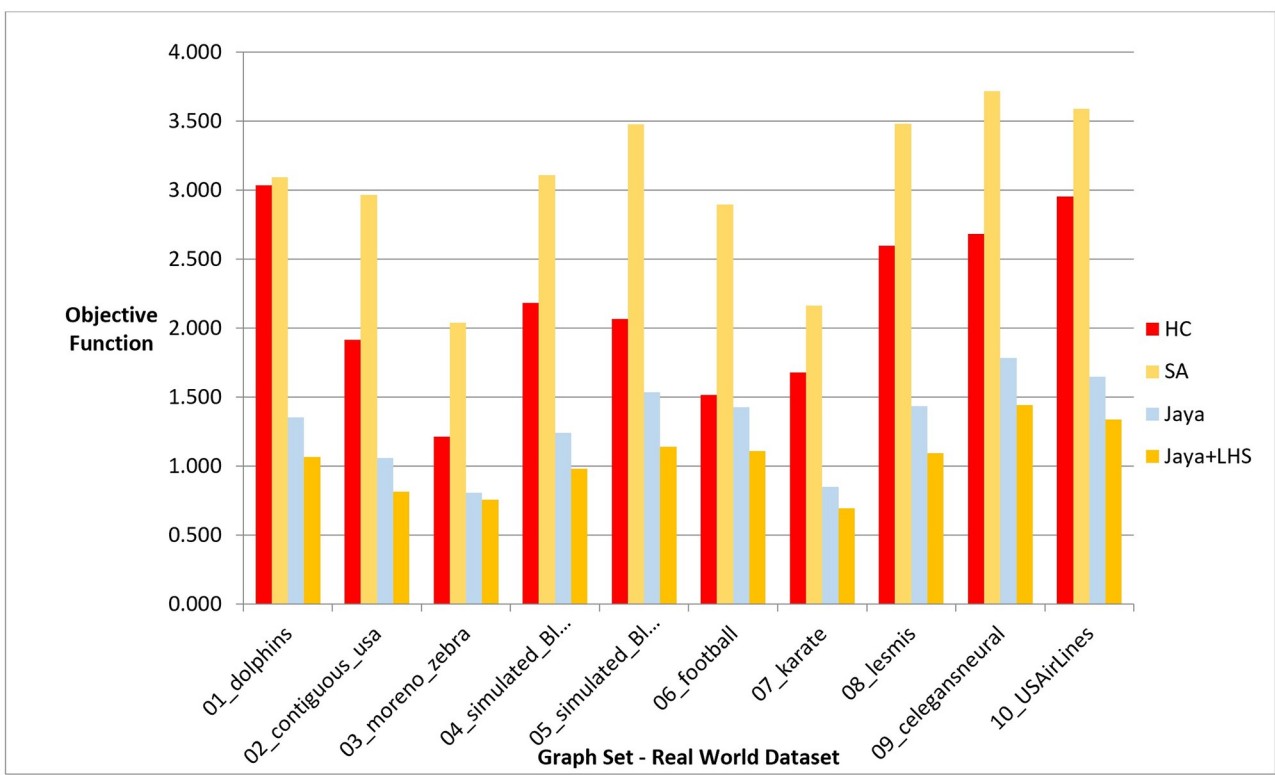

**Fig 13. A bar chart for the values of objective function on real-world dataset—Phase II.**

we conducted a Friedman test [42] and Cohen's effect size analysis [43]. The Friedman test results showed a statistically significant difference among the methods, with a p-value of 1.38E-06, comfortably below the generally accepted threshold of 0.05, thus enabling us to reject the null hypothesis that there is no difference among the methods. Cohen's effect size analysis showed that the magnitude of the difference between the pairs of methods was greater than 1, indicating a large effect size as demonstrated in Tables 16–19 for the values of the objective function and Tables 20–23 for the number of function evaluations. Overall, these results suggest that Jaya algorithm with LHS gives the best layout with a reasonable number of evaluated solutions, while Hill Climbing is faster but produces the least objective function. These results were also confirmed when applying the methods to real-world datasets described in Figs 11 and 12.

In phase II, the drawing methods were executed for a set amount of time to generate a specific number of solutions and evaluate the resulting layout quality. The results indicated that Jaya algorithm with LHS produced the best quality layouts, followed by the original Jaya algorithm, Hill Climbing, and Simulated Annealing, respectively as shown in Fig 8. The statistical significance of the results was also confirmed by the Friedman test and Cohen's effect size analysis, with a p-value of 1.38E-06 and a large effect size greater than 1 as demonstrated in Tables 24–28. These results were also supported by the real-world dataset results described in Fig 13.

In phase III, we tested the number of evaluated solutions required to achieve a certain layout quality. The results showed that Jaya algorithm with LHS required fewer evaluated solutions to generate a high-quality layout compared to the original Jaya algorithm, Hill Climbing, and Simulated Annealing as shown in Fig 9. The statistical significance of the results was

**Table 13. Summary statistics for the values of objective function on real-world dataset—Phase II.**

| Graph Set | Objective Function | | | | | | | | | | | | | | | |
|---|---|---|---|---|---|---|---|---|---|---|---|---|---|---|---|---|
| | Hill Climbing | | | | Simulated Annealing | | | | Jaya | | | | Jaya + LHS | | | |
| | Mean | Median | Max | Min | Mean | Median | Max | Min | Mean | Median | Max | Min | Mean | Median | Max | Min |
| 01_dolphins | 3.036 | 3.036 | 3.036 | 3.036 | 3.093 | 2.975 | 3.497 | 2.807 | 1.354 | 1.396 | 1.460 | 1.207 | 1.067 | 1.090 | 1.180 | 0.931 |
| 02_contiguous_usa | 1.915 | 1.915 | 1.915 | 1.915 | 2.967 | 2.959 | 3.248 | 2.694 | 1.059 | 1.099 | 1.187 | 0.889 | 0.817 | 0.822 | 0.866 | 0.763 |
| 03_moreno_zebra | 1.214 | 1.214 | 1.214 | 1.214 | 2.040 | 2.039 | 2.205 | 1.876 | 0.806 | 0.806 | 0.831 | 0.781 | 0.759 | 0.751 | 0.803 | 0.722 |
| 04_simulated_BlockModelGraph_50Nodes | 2.183 | 2.183 | 2.183 | 2.183 | 3.110 | 3.036 | 3.307 | 2.987 | 1.241 | 1.252 | 1.306 | 1.166 | 0.983 | 0.908 | 1.154 | 0.888 |
| 05_simulated_BlockModelGraph_100Nodes | 2.069 | 2.069 | 2.069 | 2.069 | 3.476 | 3.525 | 3.534 | 3.369 | 1.537 | 1.575 | 1.636 | 1.399 | 1.143 | 1.108 | 1.255 | 1.065 |
| 06_football | 1.517 | 1.517 | 1.517 | 1.517 | 2.896 | 2.823 | 3.109 | 2.754 | 1.427 | 1.450 | 1.504 | 1.327 | 1.109 | 1.154 | 1.224 | 0.949 |
| 07_karate | 1.678 | 1.678 | 1.678 | 1.678 | 2.165 | 2.164 | 2.275 | 2.057 | 0.849 | 0.914 | 0.947 | 0.687 | 0.695 | 0.689 | 0.897 | 0.500 |
| 08_lesmis | 2.597 | 2.597 | 2.597 | 2.597 | 3.481 | 3.401 | 3.741 | 3.300 | 1.437 | 1.506 | 1.614 | 1.192 | 1.096 | 1.127 | 1.243 | 0.919 |
| 09_celegansneural | 2.685 | 2.685 | 2.685 | 2.685 | 3.719 | 3.796 | 3.849 | 3.512 | 1.785 | 1.802 | 1.838 | 1.715 | 1.443 | 1.427 | 1.550 | 1.350 |
| 10_USAirLines | 2.955 | 2.955 | 2.955 | 2.955 | 3.590 | 3.513 | 3.796 | 3.462 | 1.647 | 1.527 | 1.894 | 1.521 | 1.338 | 1.366 | 1.481 | 1.168 |

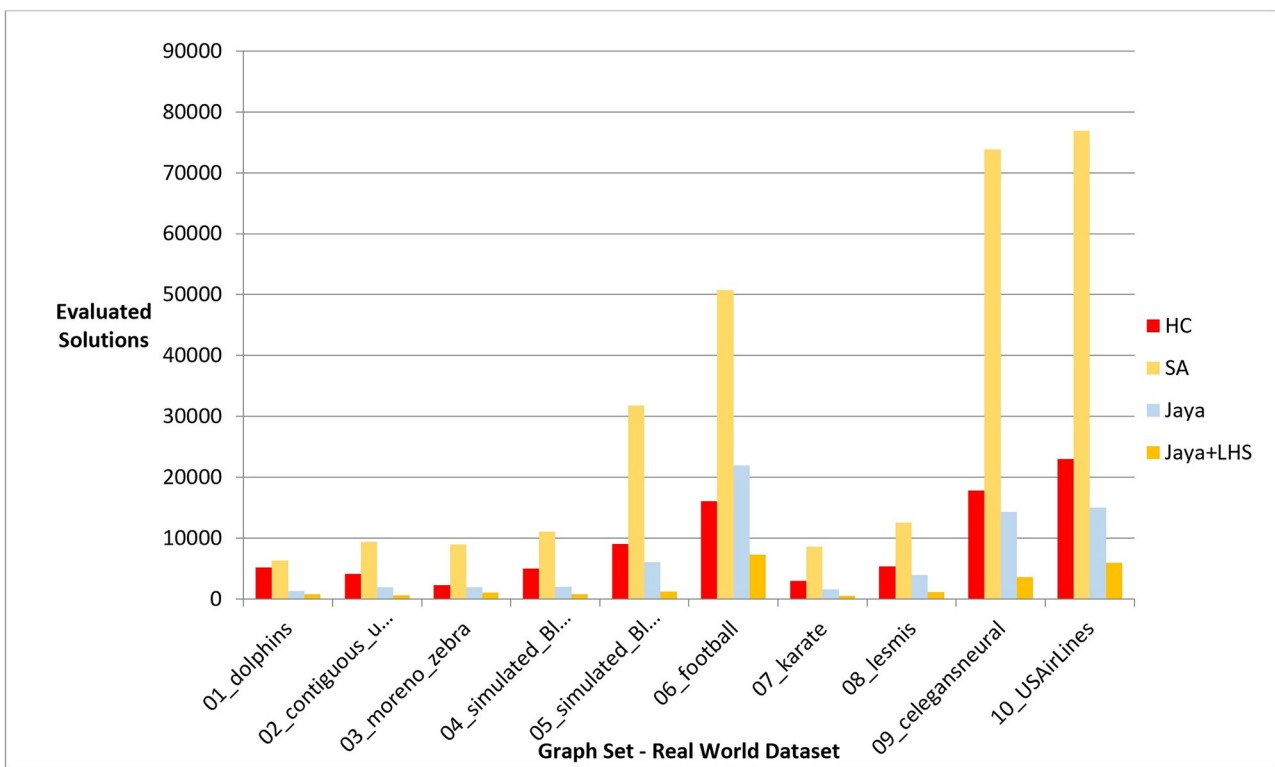

**Fig 14. A bar chart for the number of function evaluations on real-world dataset—Phase III.**

confirmed by the Friedman test and Cohen's effect size analysis, with a p-value of 1.38E-06 and a large effect size greater than 1 as described in Tables 29–33. These results were also confirmed by the real-world dataset results demonstrated in Fig 14.

The distinct characteristics of the graph drawing problem play a crucial role in justifying why Simulated Annealing evaluates more solutions than the Jaya algorithm. Graph drawing, which involves positioning nodes and edges within a confined two-dimensional space—typically a screen size or a pre-defined area—possesses a comparatively smaller solution space unlike many other optimization problems.

Given this constrained search space in graph drawing, the Jaya algorithm, as a population-based method, is capable of significantly covering and exploring this limited search space with fewer evaluations. The population in Jaya contributes to diversity and allows for simultaneous examination of various solutions, thereby aiding in uncovering optimal or near-optimal layouts within this confined search space.

Simulated Annealing as a local search algorithm, on the other hand, navigates the solution space by progressively moving to nearby solutions. Although it is proficient at escaping local optima and exploring the immediate vicinity of the current solution, the restricted search space in graph drawing may present a limited number of unique neighboring solutions. Hence, it might require more evaluations to thoroughly explore the available search space and reach convergence. Its local search mechanism primarily investigates the vicinity of a current solution, accepting worse solutions probabilistically to ensure a comprehensive exploration. This tendency explains the higher count of evaluated solutions compared to population-based methods like Jaya.

**Table 14. Summary statistics for the number of function evaluations on real-world dataset—Phase III.**

| Graph Set | Hill Climbing | | | | Simulated Annealing | | | | Jaya | | | | Jaya + LHS | | | |
|---|---|---|---|---|---|---|---|---|---|---|---|---|---|---|---|---|
| | Mean | Median | Max | Min | Mean | Median | Max | Min | Mean | Median | Max | Min | Mean | Median | Max | Min |
| 01_dolphins | 5155 | 5155 | 5155 | 5155 | 6319 | 6705 | 7907 | 4345 | 1335 | 1512 | 1713 | 579 | 768 | 813 | 1146 | 12 |
| 02_contiguous_usa | 4161 | 4161 | 4161 | 4161 | 9437 | 9210 | 10148 | 8954 | 1962 | 1812 | 2712 | 1362 | 612 | 630 | 912 | 462 |
| 03_moreno_zebra | 2292 | 2292 | 2292 | 2292 | 8985 | 8788 | 11847 | 6320 | 1944 | 2028 | 2280 | 1524 | 1104 | 1020 | 2028 | 264 |
| 04_simulated_BlockModelGraph_50Nodes | 4987 | 4987 | 4987 | 4987 | 11047 | 9941 | 13689 | 9510 | 2001 | 1389 | 4143 | 471 | 777 | 760 | 930 | 471 |
| 05_simulated_BlockModelGraph_100Nodes | 9026 | 9026 | 9026 | 9026 | 31777 | 33013 | 38619 | 23698 | 6072 | 6375 | 9102 | 2739 | 1224 | 1321 | 1830 | 921 |
| 06_football | 16108 | 16108 | 16108 | 16108 | 50759 | 55577 | 82772 | 33927 | 21936 | 21936 | 22980 | 20892 | 7320 | 7320 | 9408 | 5232 |
| 07_karate | 2958 | 2958 | 2958 | 2958 | 8576 | 7821 | 10670 | 7236 | 1587 | 957 | 3162 | 642 | 537 | 545 | 642 | 327 |
| 08_lesmis | 5342 | 5342 | 5342 | 5342 | 12525 | 12302 | 14495 | 10778 | 3990 | 4926 | 6330 | 714 | 1182 | 1091 | 1416 | 714 |
| 09_celegansneural | 17804 | 17804 | 17804 | 17804 | 73796 | 74886 | 82051 | 64450 | 14316 | 10740 | 26832 | 5376 | 3588 | 3695 | 5376 | 2694 |
| 10_USAirLines | 22974 | 22974 | 22974 | 22974 | 76901 | 76203 | 88122 | 66377 | 14997 | 17994 | 20991 | 6006 | 6006 | 6123 | 7534 | 5989 |

Evaluated Solutions

Initial layout for 03_moreno_zebra with 28 nodes and 118 edges

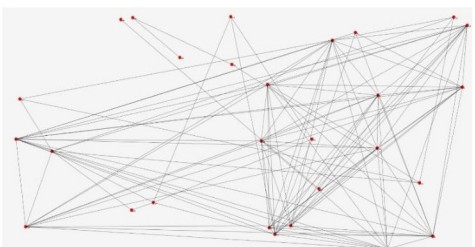

Hill Climbing layout        Simulated Annealing layout

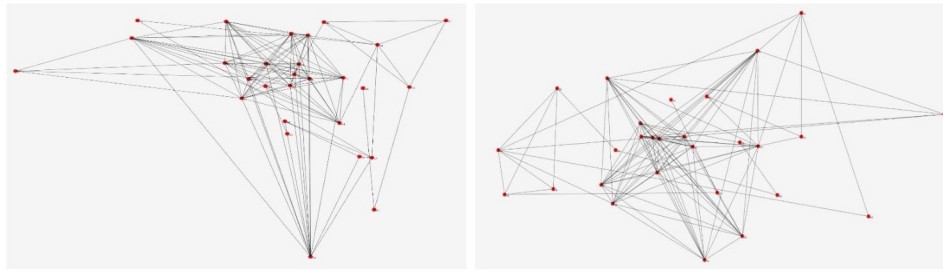

Jaya layout        Jaya with LHS layout

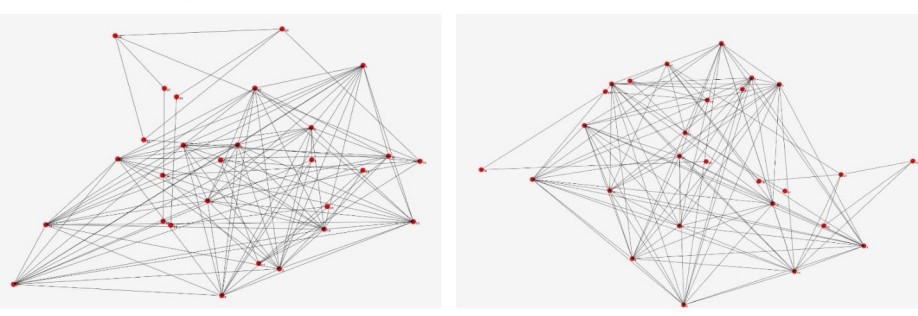

**Fig 15. Layouts of graph 3 "03_moreno_zebra" in Table 10 with 28 nodes and 118 edges.**

**Table 15. Description of the dataset used in scalability experimentation.**

| Group | Nodes | Edges |
|---|---|---|
| N50E100 | 50 | 100 |
| N100E200 | 100 | 200 |
| N150E300 | 150 | 300 |
| N200E400 | 200 | 400 |
| N250E500 | 250 | 500 |
| N300E600 | 300 | 600 |
| N350E700 | 350 | 700 |
| N400E800 | 400 | 800 |
| N450E900 | 450 | 900 |
| N500E1000 | 500 | 1000 |

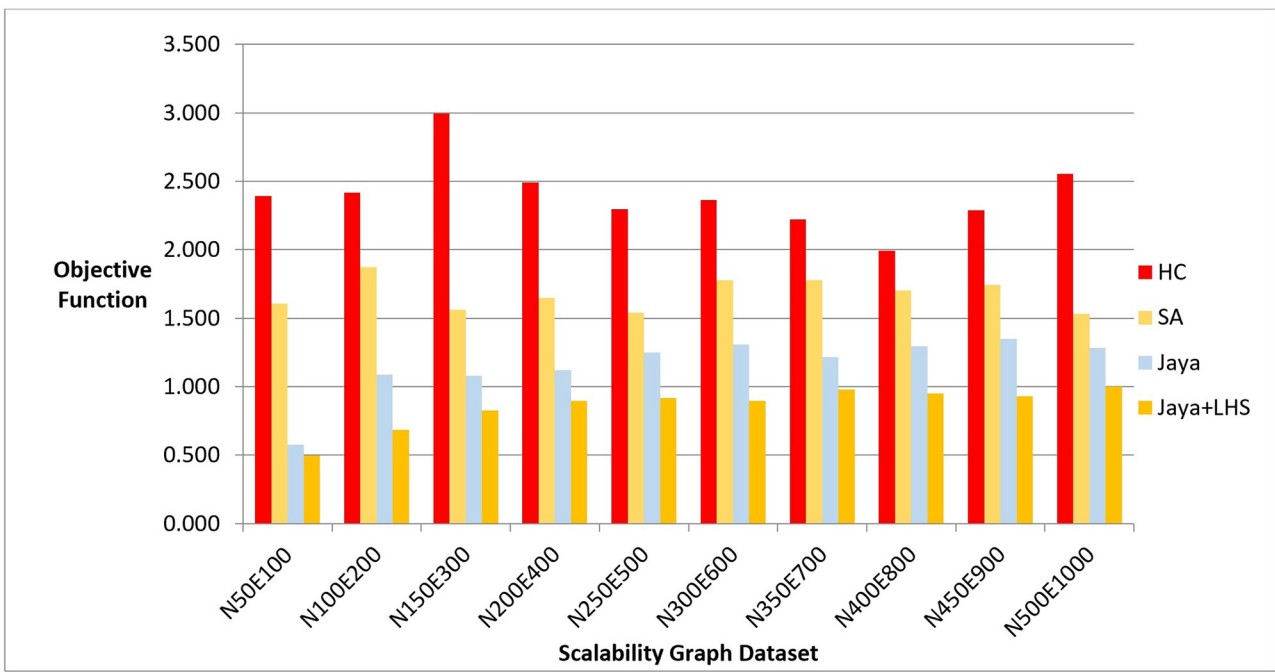

**Fig 16. A bar chart for the values of objective function on the scalability dataset.**

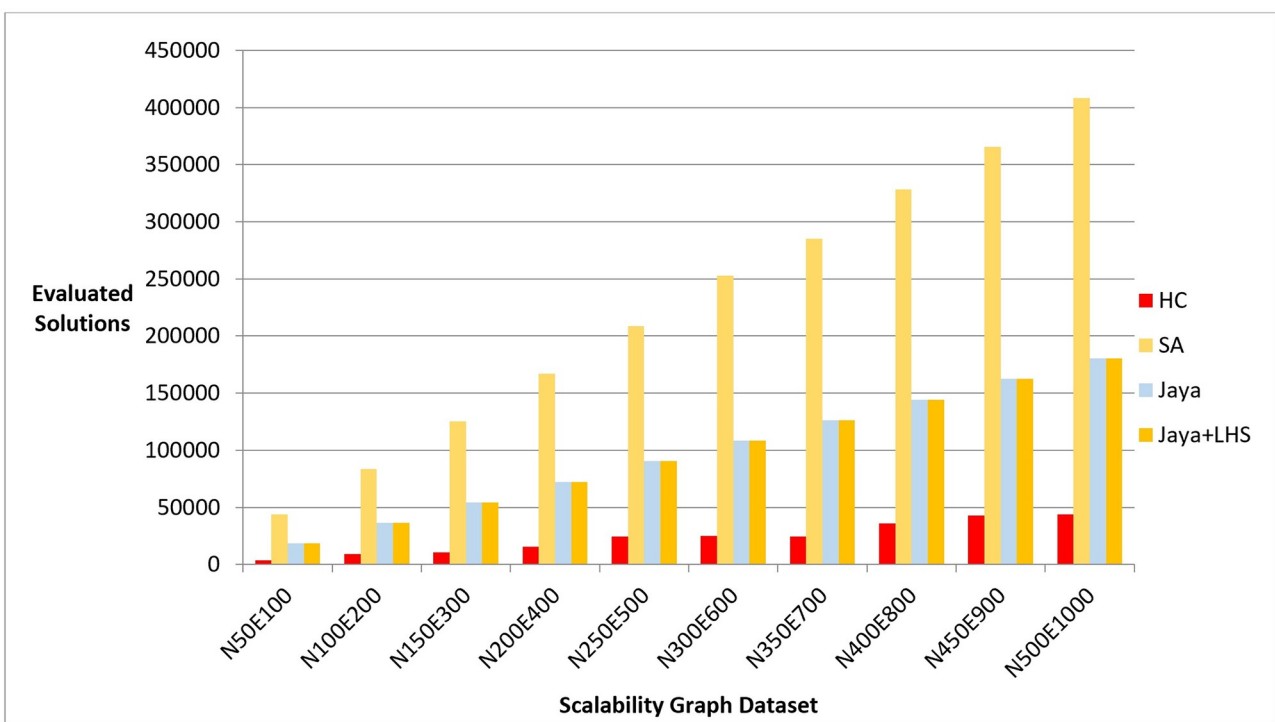

**Fig 17. A bar chart for the number of function evaluations on the scalability dataset.**

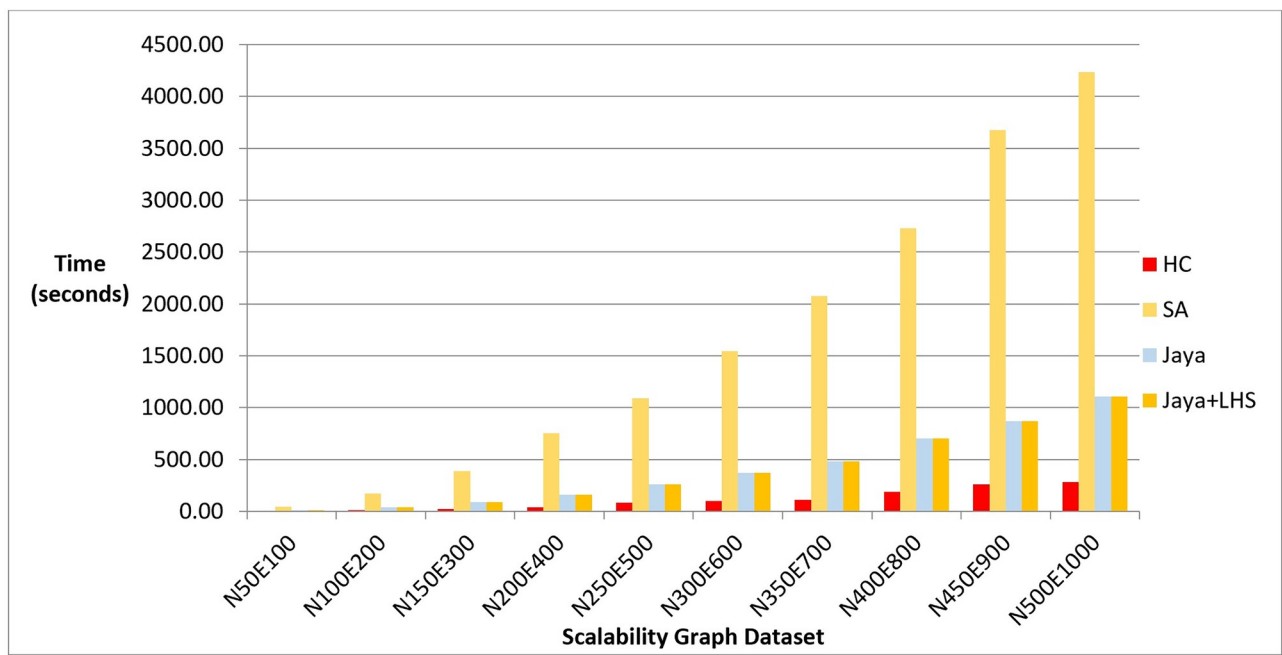

**Fig 18. A bar chart for the execution time in seconds on scalability dataset.**

In terms of scalability, Jaya algorithm with LHS was found to be the most suitable method for large graphs, as it produced high-quality layouts with reasonable evaluated solutions compared to the high number of evaluated solutions required by Simulated Annealing. Hill Climbing, on the other hand, was found to be the fastest method for large graphs, but the quality of the produced layouts was not as good as the other methods.

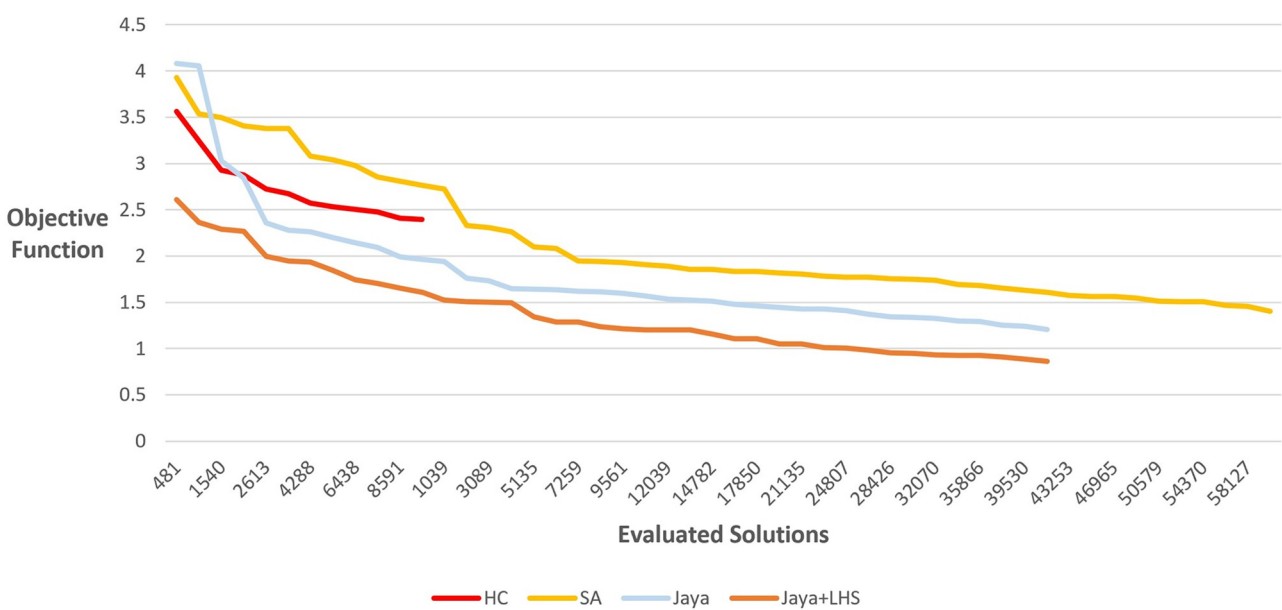

**Fig 19. Objective function variations with respect to increasing evaluations of solutions.**

**Table 16. Cohen's effect size of means of objective function on N50E100 in Table 2—Phase I.**

|  | HC | SA | Jaya | Jaya+LHS |
|---|---|---|---|---|
| HC | NA | 2.92 | 12.02 | 14.11 |
| SA | 2.92 | NA | 5.92 | 6.92 |
| Jaya | 12.02 | 5.92 | NA | 1.57 |
| Jaya+LHS | 14.11 | 6.92 | 1.57 | NA |

**Table 17. Cohen's effect size of means of objective function on N100E200 in Table 2—Phase I.**

|  | HC | SA | Jaya | Jaya+LHS |
|---|---|---|---|---|
| HC | NA | 5.65 | 8.82 | 12.22 |
| SA | 5.65 | NA | 4.70 | 10.07 |
| Jaya | 8.82 | 4.70 | NA | 4.11 |
| Jaya+LHS | 12.22 | 10.07 | 4.11 | NA |

**Table 18. Cohen's effect size of means of objective function on N150E300 in Table 2—Phase I.**

|  | HC | SA | Jaya | Jaya+LHS |
|---|---|---|---|---|
| HC | NA | 4.06 | 9.14 | 11.63 |
| SA | 4.06 | NA | 5.18 | 7.89 |
| Jaya | 9.14 | 5.18 | NA | 3.09 |
| Jaya+LHS | 11.63 | 7.89 | 3.09 | NA |

**Table 19. Cohen's effect size of means of objective function on N200E400 in Table 2—Phase I.**

|  | HC | SA | Jaya | Jaya+LHS |
|---|---|---|---|---|
| HC | NA | 1.85 | 5.69 | 8.35 |
| SA | 1.85 | NA | 6.48 | 11.56 |
| Jaya | 5.69 | 6.48 | NA | 4.18 |
| Jaya+LHS | 8.35 | 11.56 | 4.18 | NA |

**Table 20. Cohen's effect size of means of function evaluations on N50E100 in Table 2—Phase I.**

|  | HC | SA | Jaya | Jaya+LHS |
|---|---|---|---|---|
| HC | NA | 198.48 | 159.89 | 159.89 |
| SA | 198.48 | NA | 140.04 | 140.04 |
| Jaya | 159.89 | 140.04 | NA | NA |
| Jaya+LHS | 159.89 | 140.04 | NA | NA |

**Table 21. Cohen's effect size of means of function evaluations on N100E200 in Table 2—Phase I.**

|  | HC | SA | Jaya | Jaya+LHS |
|---|---|---|---|---|
| HC | NA | 245.67 | 532.70 | 532.70 |
| SA | 245.67 | NA | 152.64 | 152.64 |
| Jaya | 532.70 | 152.64 | NA | NA |
| Jaya+LHS | 532.70 | 152.64 | NA | NA |

**Table 22. Cohen's effect size of means of function evaluations on N150E300 in Table 2—Phase I.**

|          | HC      | SA      | Jaya    | Jaya+LHS |
|----------|---------|---------|---------|----------|
| HC       | NA      | 150.10  | 375.00  | 375.00   |
| SA       | 150.10  | NA      | 92.83   | 92.83    |
| Jaya     | 375.00  | 92.83   | NA      | NA       |
| Jaya+LHS | 375.00  | 92.83   | NA      | NA       |

**Table 23. Cohen's effect size of means of function evaluations on N200E400 in Table 2—Phase I.**

|          | HC       | SA      | Jaya     | Jaya+LHS |
|----------|----------|---------|----------|----------|
| HC       | NA       | 257.08  | 1162.69  | 1162.69  |
| SA       | 257.08   | NA      | 157.72   | 157.72   |
| Jaya     | 1162.69  | 157.72  | NA       | NA       |
| Jaya+LHS | 1162.69  | 157.72  | NA       | NA       |

**Table 24. Cohen's effect size of means of objective function on N30E63 in Table 6—Phase II.**

|          | HC     | SA     | Jaya   | Jaya+LHS |
|----------|--------|--------|--------|----------|
| HC       | NA     | 2.17   | 9.52   | 11.83    |
| SA       | 2.17   | NA     | 6.03   | 7.50     |
| Jaya     | 9.52   | 6.03   | NA     | 1.22     |
| Jaya+LHS | 11.83  | 7.50   | 1.22   | NA       |

**Table 25. Cohen's effect size of means of objective function on N47E81 in Table 6—Phase II.**

|          | HC     | SA     | Jaya   | Jaya+LHS |
|----------|--------|--------|--------|----------|
| HC       | NA     | 4.02   | 5.13   | 5.43     |
| SA       | 4.02   | NA     | 12.45  | 11.42    |
| Jaya     | 5.13   | 12.45  | NA     | 1.64     |
| Jaya+LHS | 5.43   | 11.42  | 1.64   | NA       |

**Table 26. Cohen's effect size of means of objective function on N90E130 in Table 6—Phase II.**

|          | HC     | SA     | Jaya   | Jaya+LHS |
|----------|--------|--------|--------|----------|
| HC       | NA     | 4.00   | 11.16  | 14.66    |
| SA       | 4.00   | NA     | 13.91  | 16.93    |
| Jaya     | 11.16  | 13.91  | NA     | 3.54     |
| Jaya+LHS | 14.66  | 16.93  | 3.54   | NA       |

**Table 27. Cohen's effect size of means of objective function on N140E200 in Table 6—Phase II.**

|          | HC     | SA     | Jaya   | Jaya+LHS |
|----------|--------|--------|--------|----------|
| HC       | NA     | 3.74   | 9.34   | 11.68    |
| SA       | 3.74   | NA     | 14.27  | 16.34    |
| Jaya     | 9.34   | 14.27  | NA     | 4.20     |
| Jaya+LHS | 11.68  | 16.34  | 4.20   | NA       |

**Table 28. Cohen's effect size of means of objective function on N175E300 in Table 6—Phase II.**

|  | HC | SA | Jaya | Jaya+LHS |
|---|---|---|---|---|
| HC | NA | 8.67 | 7.25 | 10.31 |
| SA | 8.67 | NA | 22.92 | 24.15 |
| Jaya | 7.25 | 22.92 | NA | 4.83 |
| Jaya+LHS | 10.31 | 24.15 | 4.83 | NA |

**Table 29. Cohen's effect size of means of function evaluations on N40E70 in Table 8—Phase III.**

|  | HC | SA | Jaya | Jaya+LHS |
|---|---|---|---|---|
| HC | NA | 4.89 | 2.31 | 22.95 |
| SA | 4.89 | NA | 5.23 | 8.91 |
| Jaya | 2.31 | 5.23 | NA | 2.24 |
| Jaya+LHS | 22.95 | 8.91 | 2.24 | NA |

**Table 30. Cohen's effect size of means of function evaluations on N65E95 in Table 8—Phase III.**

|  | HC | SA | Jaya | Jaya+LHS |
|---|---|---|---|---|
| HC | NA | 14.28 | 3.89 | 15.74 |
| SA | 14.28 | NA | 10.89 | 21.35 |
| Jaya | 3.89 | 10.89 | NA | 3.04 |
| Jaya+LHS | 15.74 | 21.35 | 3.04 | NA |

**Table 31. Cohen's effect size of means of function evaluations on N100E145 in Table 8—Phase III.**

|  | HC | SA | Jaya | Jaya+LHS |
|---|---|---|---|---|
| HC | NA | 6.65 | 2.03 | 22.32 |
| SA | 6.65 | NA | 6.95 | 9.16 |
| Jaya | 2.03 | 6.95 | NA | 4.58 |
| Jaya+LHS | 22.32 | 9.16 | 4.58 | NA |

**Table 32. Cohen's effect size of means of function evaluations on N160E210 in Table 8—Phase III.**

|  | HC | SA | Jaya | Jaya+LHS |
|---|---|---|---|---|
| HC | NA | 6.44 | 1.43 | 28.84 |
| SA | 6.44 | NA | 6.51 | 8.98 |
| Jaya | 1.43 | 6.51 | NA | 4.66 |
| Jaya+LHS | 28.84 | 8.98 | 4.66 | NA |

**Table 33. Cohen's effect size of means of function evaluations on N210E350 in Table 8—Phase III.**

|  | HC | SA | Jaya | Jaya+LHS |
|---|---|---|---|---|
| HC | NA | 11.12 | 1.30 | 22.89 |
| SA | 11.12 | NA | 7.49 | 14.65 |
| Jaya | 1.30 | 7.49 | NA | 1.30 |
| Jaya+LHS | 22.89 | 14.65 | 1.30 | NA |

In conclusion, the results of our experiments suggest that Jaya algorithm with LHS is the most suitable method for producing high-quality graph layouts with minimum evaluated solutions. This method outperformed Hill Climbing, Simulated Annealing, and the original Jaya algorithm in terms of scalability and quality of the produced layouts.

## 6. Conclusions and future work

In this work, we presented an innovative application of the Jaya algorithm to graph drawing, an approach yet to be explored within the field. Introducing not only the standard Jaya algorithm but also a variant enhanced by Latin Hypercube Sampling (LHS) for initial population generation, our findings established the superiority of these methods over the widely used search techniques Hill Climbing and Simulated Annealing in producing high-quality graph layouts while evaluating fewer solutions. This exceptional performance is especially notable as the Jaya algorithm, apart from being easy to apply, requires only population size and the number of iterations for tuning. It is significant to note that this is the first instance of employing the Jaya algorithm in graph drawing, thereby breaking new ground in this field.

Through this work, we have developed a robust visualization tool that facilitated our experimental testing on both synthetic and real-world graph datasets. Responding to the central research questions posed at the outset of this study, we concluded the following:

- The Jaya algorithm outperforms Hill Climbing and Simulated Annealing in graph layout optimization, with both Jaya variants consistently yielding superior layouts based on the objective function calculations while evaluating fewer solutions.

- The application of Latin Hypercube Sampling (LHS) for initializing the population does indeed enhance the performance of the original Jaya graph drawing algorithm. As our results indicate, the Jaya algorithm, when combined with LHS, consistently outpaces the original Jaya algorithm across all evaluation criteria and datasets.

Specifically, Jaya algorithm with LHS typically produces graph layouts with the best objective function value, requires the fewest number of evaluated solutions, and draws good-quality layouts with fewer evaluated solutions in comparison to the other methods. This indicates that Jaya algorithm with LHS is a promising method for drawing good-quality graphs, including larger graphs where scalability is an important factor making it suitable for real-world applications.

The robustness of our findings was further strengthened through statistical validation. Utilizing both the Friedman test and Cohen's effect size analysis, we confirmed a significant performance variance between the compared methods, a variance that was not just statistically significant but also exhibited a large effect size.

The Jaya algorithm and its variant present clear advantages. They significantly outperform conventional methods, producing superior graph layouts with fewer evaluated solutions. They are also parameter-less algorithms, which simplify their application as they require no algorithm-specific control parameters, excluding population size and number of iterations. This facilitates their adoption by researchers, making them practical and robust solutions. However, while our research heralds these methods' potential, we recognize that they may have limitations. For instance, the performance of the Jaya algorithm, even with LHS, could be contingent on the specifics of the graph drawing problem at hand, such as the size and complexity of the graph. While our tests have shown promising results, the performance in other scenarios, especially with extremely large graphs, remains to be examined.

Furthermore, in our study, we compared our proposed methods to well-established benchmark methods: Hill Climbing and Simulated Annealing, which share the important

characteristic of Jaya algorithm that they have a limited number of parameters to tune. However, owHfurther research could explore the effectiveness of our proposed approach compared with more complex population-based methods such as Genetic Algorithms, Ant Colony Optimization, Grey Wolf Optimization [44], and White Shark Optimization [45], which have a wider set of parameters to tune.

Another possible future direction for this work to explore is to test the initialization techniques discussed in [46]. The study offers an extensive examination of various methods for enhancing the effectiveness of optimizers by concentrating on their initialization approaches. The research encompasses articles released from 2000 to 2021 and sorts the initialization methods into several categories, such as random numbers, quasi-random sequences, chaos theory, probability distributions, hybrid techniques, Lévy, and others. Additionally, the research underscores the importance of exploring population initialization techniques in population-based algorithms to boost their performance.

Finally, it may be valuable to investigate the use of deep learning techniques to automatically learn features of graphs that can guide the drawing process. One way to achieve this is to use a deep reinforcement learning approach, where the algorithm learns to optimize the graph layout by interacting with the environment and receiving feedback on the quality of the layout. The proposed Jaya algorithm can also be enhanced by incorporating the learned features into the objective function, allowing the algorithm to take advantage of the learned knowledge during the search process. Overall, the integration of deep learning techniques with Jaya algorithm and other population-based methods can provide a promising direction for further improving automatic graph layout.

These directions can lead to further advancements in the field of graph drawing and can ultimately enable the visualization of increasingly complex and large-scale graphs.

## Author Contributions

**Conceptualization:** Fadi K. Dib, Peter Rodgers.

**Formal analysis:** Fadi K. Dib.

**Investigation:** Fadi K. Dib.

**Methodology:** Fadi K. Dib.

**Resources:** Fadi K. Dib.

**Software:** Fadi K. Dib.

**Supervision:** Peter Rodgers.

**Validation:** Fadi K. Dib, Peter Rodgers.

**Visualization:** Fadi K. Dib.

**Writing – original draft:** Fadi K. Dib.

**Writing – review & editing:** Fadi K. Dib, Peter Rodgers.

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
