## [Decision Letter · Decision Letter 0]

26 May 2023

PONE-D-23-13763Graph Drawing using JayaPLOS ONE

Dear Dr. Dib,

Thank you for submitting your manuscript to PLOS ONE. After careful consideration, we feel that it has merit but does not fully meet PLOS ONE’s publication criteria as it currently stands. Therefore, we invite you to submit a revised version of the manuscript that addresses the points raised during the review process.

We look forward to receiving your revised manuscript.

Kind regards,

Praveen Kumar Donta, Ph.D.

Academic Editor

PLOS ONE

Journal Requirements:

Reviewers' comments:

Reviewer's Responses to Questions

**Comments to the Author**

1. Is the manuscript technically sound, and do the data support the conclusions?

Reviewer #1: Yes

Reviewer #2: Yes

2. Has the statistical analysis been performed appropriately and rigorously? 

Reviewer #1: Yes

Reviewer #2: I Don't Know

3. Have the authors made all data underlying the findings in their manuscript fully available?

Reviewer #1: No

Reviewer #2: Yes

4. Is the manuscript presented in an intelligible fashion and written in standard English?

Reviewer #1: No

Reviewer #2: Yes

5. Review Comments to the Author

Reviewer #1: The authors introduced an enhanced variant of the Jaya algorithm for graph drawing problems. However, some significant amendments which can improve the readability of the paper and its reuse are summarized as follows:

1.Generally, I recommend revising the English writing to correct some grammar errors and typos in the document.

2.The title of this research should be comprehensive. For example, the authors used an enhanced Jaya algorithm.

3.The authors should use the “Jaya algorithm or optimizer” rather than “Jaya”.

4.The authors need to minimize the Abstract. Brief information about the datasets used for evaluation purposes should be given in the Abstract.

5.The authors compared the performance of their Jaya algorithm with two local search algorithms. This is not fair when the comparison is related to the number of evaluations. The authors need to compare the performance of their Jaya algorithm with other population-based algorithms like grey wolf optimization, and White Shark Optimizer.

6.In the Introduction, the authors need to justify the researchers used optimization algorithms for graph drawing.

7.The motivations and contributions of this research are not clear. In addition, the contributions of this research should be listed as points in the Introduction Section.

8.The authors need to justify why they are using the Jaya optimizer rather than other optimization algorithms for graph drawing.

9.The most relevant research papers should be cited in the current work like "An intensive and comprehensive overview of JAYA algorithm, its versions and applications." Archives of Computational Methods in Engineering 29.2 (2022): 763-792. "Binary JAYA algorithm with adaptive mutation for feature selection." Arabian Journal for Science and Engineering 45.12 (2020): 10875-10890.

10.The authors need to add some lines at the end of the literature review section to highlight the research gap.

11.The background section related to the literature review, Jaya algorithm, and Latin Hypercube Sampling. I see this section should divide into three subsections. The authors should explain the procedural steps of the Jaya algorithm, and Latin Hypercube Sampling in this section.

12.Why do the authors in the proposed method talk about hill climbing and simulated annealing? These two optimizers are not part of the proposed algorithm. The implementation of these two algorithms should be given in the experimental section.

13.The pseudo-code given in Figures 1, 2, and 3 should be an Algorithm rather than a figure.

14.Some Figures are not clear like Figure 4.

15.I ask the authors to provide the mathematical formulation of the graph drawing problem. This includes the objective function.

16.The authors need to justify why the number of evaluated solutions in SA is higher than those of their Jaya algorithm, although the Jaya algorithm is a population algorithm and SA is a local search algorithm.

17.The convergence and population diversity of their algorithm against the classical Jaya and other algorithms should be studied.

18.The advantages and limitations of the proposed algorithm should be given in the conclusion section.

19.Conclusion is not at all satisfactory. It should be concise and your contribution and novelty should be claimed.

Reviewer #2: The authors present a detailed investigation of two versions of Jaya algorithms for automatic graph layout and compare their performances with two well-known search algorithms that are frequently used in graph drawing. The experiments were conducted on both synthetic and real-world graph datasets to evaluate the performance of the proposed methods in terms of the quality of the generated layouts and the number of function evaluations.

The paper gives a detailed description of the various experiments that were performed and the justification for the various phases of the experiments in consonants with the set out aim and objectives to answer the research questions. The paper is well written, well organized, easy to read and understand. I believe the paper merits publication. However, I recommend minor revisions as highlighted below:

1.The abstract need to be rewritten to include a brief statement to introduce automatic graph drawing and its importance, the current challenge and how the experiment address the challenge.

2.The main contributions of the paper will be better presented as a list.

3.The statistical result for the Friedman test should also be included.

4.No experimental result table is shown in the paper to justify the various resultant bar graphs presented.

6. PLOS authors have the option to publish the peer review history of their article (what does this mean?). If published, this will include your full peer review and any attached files.

Reviewer #1: No

Reviewer #2: No

---

## [Author Response · Author response to Decision Letter 0]

8 Jun 2023

We are pleased to resubmit the revised version of our paper. We appreciate the constructive feedback of the academic editor and the reviewers. We have addressed each of their concerns as outlined below in italic bold. 

PONE-D-23-13763

Graph Drawing using Jaya

PLOS ONE

Dear Dr. Dib,

Thank you for submitting your manuscript to PLOS ONE. After careful consideration, we feel that it has merit but does not fully meet PLOS ONE’s publication criteria as it currently stands. Therefore, we invite you to submit a revised version of the manuscript that addresses the points raised during the review process.

We look forward to receiving your revised manuscript.

Kind regards,

Praveen Kumar Donta, Ph.D.

Academic Editor

PLOS ONE

Journal Requirements:

• Style requirements and figure files naming were revised. 

• Reference list was revised. None of the references has been retracted. Also, few references were also added to address the reviewers’ comments: 

Mirjalili S, Mirjalili SM, Lewis A. Grey wolf optimizer. Adv Eng Softw. 2014;69:46–61. 

Braik M, Hammouri A, Atwan J, Al-Betar MA, Awadallah MA. White Shark Optimizer: A novel bio-inspired meta-heuristic algorithm for global optimization problems. Knowl-Based Syst. 2022;243:108457.

Reviewers' comments:

Reviewer's Responses to Questions

Comments to the Author

1. Is the manuscript technically sound, and do the data support the conclusions?

Reviewer #1: Yes

Reviewer #2: Yes

2. Has the statistical analysis been performed appropriately and rigorously?

Reviewer #1: Yes

Reviewer #2: I Don't Know

3. Have the authors made all data underlying the findings in their manuscript fully available?

Reviewer #1: No

Reviewer #2: Yes

4. Is the manuscript presented in an intelligible fashion and written in standard English?

Reviewer #1: No

Reviewer #2: Yes

5. Review Comments to the Author

Reviewer #1: The authors introduced an enhanced variant of the Jaya algorithm for graph drawing problems. However, some significant amendments which can improve the readability of the paper and its reuse are summarized as follows:

1.Generally, I recommend revising the English writing to correct some grammar errors and typos in the document.

• Revised. 

2.The title of this research should be comprehensive. For example, the authors used an enhanced Jaya algorithm.

• We fully understand this point and we appreciate the effort to make our research's novelty more visible. However, we believe that the current title, "Graph drawing using Jaya", can be more beneficial for the wider research community. The primary reason behind this decision is the potential discoverability of the paper. As this is the first application of the Jaya algorithm in the field of graph drawing, we anticipate that researchers interested in novel applications of this algorithm might use "Jaya" and "graph drawing" as keywords in their searches. We feel that the current title, which contains both of these key terms, would ensure that our paper appears in such searches and is accessible to a wider audience. It also reflects the presentation in the paper where we compare a standard Jaya algorithm with an enhanced Jaya algorithm. Additionally, the enhancements we've introduced to the Jaya algorithm are detailed within the paper, providing readers with a clear understanding of our contributions. We agree that the enhancements are significant, and we believe the abstract and the paper content adequately highlight this. 

Having said that, we are open to change if the editorial board still feels that the title is not appropriate.

3.The authors should use the “Jaya algorithm or optimizer” rather than “Jaya”.

• “Jaya algorithm” is used in the revised version. 

4.The authors need to minimize the Abstract. Brief information about the datasets used for evaluation purposes should be given in the Abstract.

• The abstract is slightly minimized. Brief information on the synthetic dataset and introduction to automatic graph drawing and its importance are added in the revised version to address the comments of all reviewers. 

5.The authors compared the performance of their Jaya algorithm with two local search algorithms. This is not fair when the comparison is related to the number of evaluations. The authors need to compare the performance of their Jaya algorithm with other population-based algorithms like grey wolf optimization, and White Shark Optimizer.

• In the proposed method section, we clarify the rationale behind our comparison against hill climbing and simulated annealing: “The rationale behind this decision is to evaluate the performance of Jaya algorithm against search algorithms for graph drawing that also have few parameters to tune. We also aim to assess the ability of Jaya algorithm to adapt to the complexities of the graph layout problem, as well as its performance in terms of convergence rate, solution quality, and computational efficiency. By comparing these algorithms, we can demonstrate the effectiveness and efficiency of methods with low numbers of parameters, encouraging other researchers in the field to consider using such approaches. This comparison not only highlights their relative performance but also emphasizes the benefits of using algorithms that require less tuning, allowing practitioners to focus more on solving the graph layout problem instead of spending significant time and resources fine-tuning algorithm parameters. 

Additionally, the reduced need for parameter tuning makes these methods more robust and generalizable across various graph layout problems. This can lead to more consistent performance and better reproducibility in research.” 

In addition to that, Grey Wolf Optimization (GWO) and White Shark Optimization (WSO) require several parameters to tune, such as the coefficients that control the movement of the wolves and factors that control the convergence of the wolves in GWO, and chaotic and inclination factors in WSO in addition to the maximum number of iterations, population size, and step size. However, we believe that this can be investigated in further research that focuses on the most effective population-based method in the field of graph drawing as we mention in the future work section: “Our proposed Jaya algorithm with LHS approach was compared to well-established benchmark methods: Hill Climbing and Simulated Annealing, which share the important characteristic of Jaya algorithm that they have a limited number of parameters to tune. Further research could explore the effectiveness of our proposed approach compared with more complex population-based methods such as Genetic Algorithms and Ant Colony Optimization, which have a wider set of parameters to tune.” 

In the revised version of this paper, we modified that paragraph to include GWO and WSO along with their citations to become as follows: “Our proposed Jaya algorithm with LHS approach was compared to well-established benchmark methods: Hill Climbing and Simulated Annealing, which share the important characteristic of Jaya algorithm that they have a limited number of parameters to tune. Further research could explore the effectiveness of our proposed approach compared with more complex population-based methods such as Genetic Algorithms, Ant Colony Optimization, Grey Wolf Optimization [44], and White Shark Optimization [45], which have a wider set of parameters to tune.” 

6.In the Introduction, the authors need to justify the researchers used optimization algorithms for graph drawing.

• The following paragraph in the introduction explains the rationale behind using optimization by researchers in the field of graph drawing: “On the other hand, search-based methods search the solution space by generating a sequence of candidate solutions and iteratively improving them based on a predefined objective function. Search-based methods have gained popularity in graph drawing due to their ability to handle multi-objective functions. These methods can combine multiple quality metrics into a single function to be optimized.” 

Also we list a good number of citations for researchers who used optimization in the field of graph drawing such as: 

Davidson R, Harel D. Drawing graphs nicely using simulated annealing. ACM Trans Graph TOG. 1996;15(4):301–31.

Rosete-Suárez A, Ochoa-Rodríguez A, Sebag M. Automatic graph drawing and stochastic hill-climbing. In: Genetic and Evolutionary Computation Conference. Morgan Kaufmann; 1999. p. 1699–706. 

Gibson H, Faith J, Vickers P. A survey of two-dimensional graph layout techniques for information visualisation. Inf Vis. 2013;12(3–4):324–57.

Wybrow M, Rodgers P, Dib FK. Euler diagrams drawn with ellipses area-proportionally (Edeap). BMC Bioinformatics. 2021;22:1–27. 

Dib FK, Rodgers P. Graph drawing using tabu search coupled with path relinking. PloS One. 2018;13(5):e0197103.

Stott J, Rodgers P, Martinez-Ovando JC, Walker SG. Automatic metro map layout using multicriteria optimization. IEEE Trans Vis Comput Graph. 2010;17(1):101–14.

7.The motivations and contributions of this research are not clear. In addition, the contributions of this research should be listed as points in the Introduction Section.

• In the introduction, we demonstrate our motivation in the following paragraphs: 

“In this research, we propose the Jaya algorithm method [8] for automatic graph layout to improve the efficiency and effectiveness of drawing general graph layouts with straight lines based on a weighted sum multi-criteria optimization. To our knowledge, Jaya algorithm has never been used in the field of graph drawing. Jaya algorithm is a population-based search method that maintains a population of candidate solutions, where solutions are updated based on the best solutions found in the population. We introduce Jaya algorithm as it has no algorithm-specific control parameters other than population size and number of iterations, which makes it easy for researchers to apply in the field, and it has been proven effective in many applications [9,10].

Our study aimed to answer two main research questions. Firstly, does Jaya algorithm perform better than Hill Climbing and Simulated Annealing approaches in the field of graph layout? Secondly, does applying Latin Hypercube Sampling (LHS) for initializing the population for Jaya algorithm improve the performance of the original Jaya graph drawing algorithm method? To answer these questions, we implemented and evaluated these methods alongside Hill Climbing and Simulated Annealing. We conducted three types of evaluations: finding the best layout achievable by minimizing the value of the objective function, measuring the quality of the graph layout after a fixed optimization time (number of function evaluations), and determining the speed to draw an acceptable layout. We compared Jaya algorithm with Hill Climbing and Simulated Annealing, as they have fewer parameters to tune than other search methods.”

• We added the contributions as a list of points in the introduction:

“The main contributions of this research can be summarized as follows: 

• Our study pioneers the use of the Jaya algorithm in graph drawing, outperforming conventional search optimization methods significantly. It is a parameter-less algorithm, requiring no algorithm-specific control parameters, making it easy for researchers to apply in the field.

• We optimized the Jaya algorithm by integrating the Latin Hypercube Sampling (LHS) method for population initialization, boosting its overall efficiency.

• We developed an intuitive visualization tool that facilitates the evaluation and comparison of different optimization techniques for graph layout by researchers and practitioners.”

8.The authors need to justify why they are using the Jaya optimizer rather than other optimization algorithms for graph drawing.

• In the introduction, we justify the reason in two paragraphs: “Researchers in the field of multi-criteria graph drawing benefit from methods that are easy to implement and which require few parameters to tune. This allows researchers and practitioners to focus on solving the graph layout problem rather than spending significant time and effort fine-tuning algorithm parameters [5].” And “We introduce Jaya algorithm as it has no algorithm-specific control parameters other than population size and number of iterations, which makes it easy for researchers to apply in the field, and it has been proven effective in many applications [9,10].” 

9.The most relevant research papers should be cited in the current work like "An intensive and comprehensive overview of JAYA algorithm, its versions and applications." Archives of Computational Methods in Engineering 29.2 (2022): 763-792. "Binary JAYA algorithm with adaptive mutation for feature selection." Arabian Journal for Science and Engineering 45.12 (2020): 10875-10890.

• Both papers have already been cited in our manuscript [10] and [21]. 

10.The authors need to add some lines at the end of the literature review section to highlight the research gap.

• The following paragraph has been added to the paper: “The current state of the art in search-based graph drawing methods have a number of drawbacks leading to a research gap. Techniques such as Simulated Annealing, Hill Climbing, and Genetic Algorithms often struggle with suboptimal solutions, slow rates of convergence, and the complexity of parameter tuning. Additionally, the initial population choice in can significantly impact the performance and convergence of the optimization process, potentially leading to premature convergence or stagnation. Although some research has addressed parameter tuning as a bi-objective process, and various methods have been proposed for efficient population initialization, such as Latin Hypercube Sampling (LHS), these techniques have not been combined in a comprehensive approach to graph drawing. Moreover, the potential of the Jaya algorithm remains unexplored in the field of graph drawing. This research aims to address these gaps by pioneering the use of the Jaya algorithm in graph drawing, integrating LHS for efficient population initialization, and facilitating the evaluation and comparison of different optimization techniques for graph layout through a user-friendly visualization tool.” 

11.The background section related to the literature review, Jaya algorithm, and Latin Hypercube Sampling. I see this section should divide into three subsections. The authors should explain the procedural steps of the Jaya algorithm, and Latin Hypercube Sampling in this section.

• The background section has been divided into three sections as recommended: 

2.1 Search-Based Optimization Methods for Graph Drawing

2.2 The Jaya Algorithm and the Importance of Parameter Tuning

2.3 Latin Hypercube Sampling in Optimization Algorithms

Regarding the explanation of the procedural steps of the Jaya algorithm and LHS, we have chosen to present these procedures in the proposed method section because they represent our unique approach and implementation in the field of graph drawing, which is separate from the broader context provided in the background section. We believe that this structure would allow our readers to distinguish between the established techniques and our novel application of them more effectively.

12.Why do the authors in the proposed method talk about hill climbing and simulated annealing? These two optimizers are not part of the proposed algorithm. The implementation of these two algorithms should be given in the experimental section.

• We moved the implementation of hill climbing and simulated annealing from the proposed method section to the experimental results section. 

13.The pseudo-code given in Figures 1, 2, and 3 should be an Algorithm rather than a figure.

• The algorithms have been included directly in the text of the paper, rather than being presented as separate figures. 

14.Some Figures are not clear like Figure 4.

• Figure 4 (now Figure 1) has been changed to match PLOS ONE recommended guidelines using https://pacev2.apexcovantage.com/

15.I ask the authors to provide the mathematical formulation of the graph drawing problem. This includes the objective function.

• The mathematical formulation of the metrics and the objective function have been added to the proposed method section. 

16.The authors need to justify why the number of evaluated solutions in SA is higher than those of their Jaya algorithm, although the Jaya algorithm is a population algorithm and SA is a local search algorithm.

• The following justification has been added to analysis of results section: “The distinct characteristics of the graph drawing problem play a crucial role in justifying why Simulated Annealing evaluates more solutions than the Jaya algorithm. Graph drawing, which involves positioning nodes and edges within a confined two-dimensional space - typically a screen size or a pre-defined area - possesses a comparatively smaller solution space unlike many other optimization problems.

Given this constrained search space in graph drawing, the Jaya algorithm, as a population-based method, is capable of significantly covering and exploring this limited search space with fewer evaluations. The population in Jaya contributes to diversity and allows for simultaneous examination of various solutions, thereby aiding in uncovering optimal or near-optimal layouts within this confined search space.

Simulated Annealing as a local search algorithm, on the other hand, navigates the solution space by progressively moving to nearby solutions. Although it is proficient at escaping local optima and exploring the immediate vicinity of the current solution, the restricted search space in graph drawing may present a limited number of unique neighboring solutions. Hence, it might require more evaluations to thoroughly explore the available search space and reach convergence. Its local search mechanism primarily investigates the vicinity of a current solution, accepting worse solutions probabilistically to ensure a comprehensive exploration. This tendency explains the higher count of evaluated solutions compared to population-based methods like Jaya.”

17.The convergence and population diversity of their algorithm against the classical Jaya and other algorithms should be studied.

• Section 4.6 (Performance) has been added including a figure (Fig. 19) that demonstrates objective function variations with respect to increasing evaluations of solutions.

18.The advantages and limitations of the proposed algorithm should be given in the conclusion section.

• We have restructured the conclusion section to incorporate a discussion of both the strengths and constraints of our approach. 

19.Conclusion is not at all satisfactory. It should be concise and your contribution and novelty should be claimed.

• We have restructured the conclusion section. 

Reviewer #2: The authors present a detailed investigation of two versions of Jaya algorithms for automatic graph layout and compare their performances with two well-known search algorithms that are frequently used in graph drawing. The experiments were conducted on both synthetic and real-world graph datasets to evaluate the performance of the proposed methods in terms of the quality of the generated layouts and the number of function evaluations.

The paper gives a detailed description of the various experiments that were performed and the justification for the various phases of the experiments in consonants with the set out aim and objectives to answer the research questions. The paper is well written, well organized, easy to read and understand. I believe the paper merits publication. However, I recommend minor revisions as highlighted below:

1.The abstract need to be rewritten to include a brief statement to introduce automatic graph drawing and its importance, the current challenge and how the experiment address the challenge.

• The abstract is slightly minimized. Brief information on the synthetic dataset and introduction to automatic graph drawing and its importance are added in the revised version to address the comments of all reviewers. 

2.The main contributions of the paper will be better presented as a list.

• We added the contributions as a list of points in the introduction:

“The main contributions of this research can be summarized as follows: 

• Our study pioneers the use of the Jaya algorithm in graph drawing, outperforming conventional search optimization methods significantly. It is a parameter-less algorithm, requiring no algorithm-specific control parameters, making it easy for researchers to apply in the field.

• We optimized the Jaya algorithm by integrating the Latin Hypercube Sampling (LHS) method for population initialization, boosting its overall efficiency.

• We developed an intuitive visualization tool that facilitates the evaluation and comparison of different optimization techniques for graph layout by researchers and practitioners.”

3.The statistical result for the Friedman test should also be included.

• The statistical results of Friedman test are already integrated within the text in the analysis of results section. In addition to that, this statement has been added “The Friedman test results showed a statistically significant difference among the methods, with a p-value of 1.38E-06, comfortably below the generally accepted threshold of 0.05, thus enabling us to reject the null hypothesis that there is no difference among the methods.” 

4.No experimental result table is shown in the paper to justify the various resultant bar graphs presented.

• The experimental result tables that correspond to the bar charts have been added to the experimental results section for both random and real world datasets. 

6. PLOS authors have the option to publish the peer review history of their article (what does this mean?). If published, this will include your full peer review and any attached files.

Do you want your identity to be public for this peer review? For information about this choice, including consent withdrawal, please see our Privacy Policy.

Reviewer #1: No

Reviewer #2: No

---

## [Decision Letter · Decision Letter 1]

13 Jun 2023

Graph Drawing using Jaya

PONE-D-23-13763R1

Dear Dr. Dib,

We’re pleased to inform you that your manuscript has been judged scientifically suitable for publication and will be formally accepted for publication once it meets all outstanding technical requirements.

Kind regards,

Praveen Kumar Donta, Ph.D.

Academic Editor

PLOS ONE

Additional Editor Comments (optional):

Reviewers' comments:

Reviewer's Responses to Questions

**Comments to the Author**

1. If the authors have adequately addressed your comments raised in a previous round of review and you feel that this manuscript is now acceptable for publication, you may indicate that here to bypass the “Comments to the Author” section, enter your conflict of interest statement in the “Confidential to Editor” section, and submit your "Accept" recommendation.

Reviewer #1: All comments have been addressed

Reviewer #2: All comments have been addressed

2. Is the manuscript technically sound, and do the data support the conclusions?

Reviewer #1: Yes

Reviewer #2: (No Response)

3. Has the statistical analysis been performed appropriately and rigorously? 

Reviewer #1: Yes

Reviewer #2: (No Response)

4. Have the authors made all data underlying the findings in their manuscript fully available?

Reviewer #1: Yes

Reviewer #2: (No Response)

5. Is the manuscript presented in an intelligible fashion and written in standard English?

Reviewer #1: Yes

Reviewer #2: (No Response)

6. Review Comments to the Author

Reviewer #1: No further comments are required. This manuscript should be accepted in its current form. Please accept it.

Reviewer #2: (No Response)

7. PLOS authors have the option to publish the peer review history of their article (what does this mean?). If published, this will include your full peer review and any attached files.

Reviewer #1: No

Reviewer #2: No

---

## [Editor Report · Acceptance letter]

19 Jun 2023

PONE-D-23-13763R1 

Graph Drawing using Jaya 

Dear Dr. Dib:

I'm pleased to inform you that your manuscript has been deemed suitable for publication in PLOS ONE. Congratulations! Your manuscript is now with our production department. 

Kind regards, 

on behalf of

Dr. Praveen Kumar Donta 

Academic Editor

PLOS ONE